# Infinite-Resolution Integral Noise Warping for Diffusion Models

**Yitong Deng**[1,2]**, Winnie Lin**[1]**, Lingxiao Li**[1]**, Dmitriy Smirnov**[1]**, Ryan Burgert**[3,4]**, Ning Yu**[3]**,
Vincent Dedun**[1]**, Mohammad H. Taghavi**[1]

[1]Netflix, [2]Stanford University, [3]Netflix Eyeline Studios, [4]Stony Brook University
yitongd@stanford.edu
{winniel, lingxiaol, dimas, vdedun, mtaghavi}@netflix.com
rburgert@cs.stonybrook.edu
ning.yu@scanlinevfx.com

## Abstract

Adapting pretrained image-based diffusion models to generate temporally consistent videos has become an impactful generative modeling research direction. Training-free noise-space manipulation has proven to be an effective technique, where the challenge is to preserve the Gaussian white noise distribution while adding in temporal consistency. Recently, Chang et al. (2024) formulated this problem using an integral noise representation with distribution-preserving guarantees, and proposed an upsampling-based algorithm to compute it. However, while their mathematical formulation is advantageous, the algorithm incurs a high computational cost. Through analyzing the limiting-case behavior of their algorithm as the upsampling resolution goes to infinity, we develop an alternative algorithm that, by gathering increments of multiple Brownian bridges, achieves their infinite-resolution accuracy while simultaneously reducing the computational cost by orders of magnitude. We prove and experimentally validate our theoretical claims, and demonstrate our method's effectiveness in real-world applications. We further show that our method readily extends to the 3-dimensional space.

## 1 Introduction

The success of diffusion models in image generation and editing (Rombach et al., 2022; Nichol et al., 2021; Ho et al., 2020; Zhang et al., 2023a) has spurred significant interest in lifting these capacities to the video domain (Singer et al., 2022; Durrett, 2019; Gupta et al., 2023; Blattmann et al., 2023; Ho et al., 2022; Guo et al., 2024). While training video diffusion models directly on spatiotemporal data is a natural idea, practical concerns such as limited availability of large-scale video data and high computational cost have motivated investigations into training-free alternatives. One such approach is to use pre-trained image models to directly generate video frames, and utilize techniques such as cross-frame attention, feature injection and hierarchical sampling to promote temporal consistency across frames (Ceylan et al., 2023; Zhang et al., 2023b; Khachatryan et al., 2023; Cong et al., 2023).

Among these techniques, the controlled initialization of noise has been consistently shown to be an important one (Ceylan et al., 2023; Khachatryan et al., 2023; Cai et al., 2024). However, most existing approaches for noise manipulation either compromise the noise Gaussianity (which introduces a domain gap at inference time), or are restricted to simple manipulations such as filtering and blending which are insufficient for capturing complex temporal correlations. Recently, Chang et al. (2024) proposed a method that both preserves Gaussian white noise distribution and captures temporal correlations via *integral noise warping*: each warped noise pixel integrates a continuous noise field over a polygonal deformed pixel region, which is computed by summing subpixels of an upsampled noise image. However, this method's theoretical soundness and effectiveness are ensued by its demanding computational cost in both memory and time, which not only incurs a significant overhead at inference time but also limits its useability in novel applications (Kwak et al., 2024).

In this paper, we introduce a new noise-warping algorithm that drastically cuts down the cost of Chang et al. (2024) while fully retaining its virtues. Our key insight for achieving this is that, when adopting an Eulerian perspective (as opposed to the original Lagrangian one), the limiting-

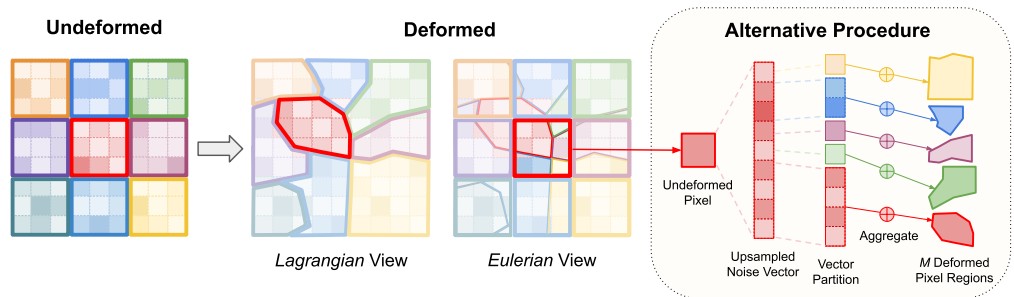

Figure 1: When the image grid deforms, the Lagrangian view tracks a deformed pixel region, while the Eulerian view tracks the undeformed pixel square as it gets partitioned into multiple regions. On the right, we leverage the exchangeability of upsampled subpixels to convert the Lagrangian gathering procedure into scattering noise subpixels to overlapped deformed pixel regions.

case algorithm of Chang et al. (2024) for computing a warped noise pixel reduces to summing over increments from multiple Brownian bridges (Durrett, 2019, Section 8.4). In place of the costly upsampling procedure, sampling the increments of a Brownian bridge can be done efficiently in an autoregressive manner (2). We build upon this to devise the *infinite-resolution integral noise warping* algorithm (1) which directly resolves noise transport in the continuous space, when given an oracle that returns the overlapping area between a pixel square and a deformed pixel region (Section 2.2).

We propose two concrete ways to compute this oracle, leading to a *grid-based* and a *particle-based* variant of our method. Similar to Chang et al. (2024), the *grid-based* variant (Algorithm 2) computes the area by explicitly constructing per-pixel deformed polygons, and is exactly equivalent to the existing approach (Chang et al., 2024) with an infinite upsampling resolution, while running $8.0\times$ to $19.7\times$ faster and using $9.22\times$ less memory[1]. Inspired by hybrid Eulerian-Lagrangian fluid simulation (Brackbill et al., 1988), our novel *particle-based* variant (Algorithm 3) computes area in a fuzzy manner, which not only offers a *further* $5.21\times$ speed-up *over our grid-based variant*, but is also agnostic to non-injective maps. In real-world scenarios, the particle-based variant shows no compromise in generation quality compared to the grid-based one (see video results), while offering superior robustness, efficiency, simplicity, and extensibility to higher dimensions.

In summary, we propose a new noise-warping method to facilitate video generation by lifting image diffusion models. Through analyzing the limiting case of the current state-of-the-art method (Chang et al., 2024) with an infinite upsampling resolution, we derive its continuous-space analogy, which fully retains its distribution-preserving and temporally coherent properties, while achieving orders-of-magnitude speed-up, warping $1024\times1024$ noise images in $\sim 0.045$s (grid variant) and $\sim 0.0086$s (particle variant) using a laptop with an Nvidia RTX 3070 Ti GPU.

## 2 METHODOLOGY

In this section, we introduce our method as follows:

- We present an equivalent Eulerian interpretation (Figure 1) of the method by Chang et al. (2024), which was originally described from a Lagrangian viewpoint.

- We show that the limiting algorithm of the Eulerian formulation as the upsampling level goes to infinity is equivalent to sampling increments of Brownian bridges.

- We present our main algorithm (1) which, given a partition record that returns the overlapping area between a pixel square and a deformed pixel region, samples increments of Brownian bridges and scatters the increments to form the warped noise image.

- We propose two concrete algorithms for computing the overlap areas. The *grid-based* Algorithm 2 extends Chang et al. (2024) to infinite resolution without the overhead of upsampling. The *particle-based* Algorithm 3 departs from grid-based discretization and uses particles instead, resulting in a simpler algorithm that is robust to degenerate maps.

---

[1]Since the official code of Chang et al. (2024) is not available, performance is compared using our reimplementation in Taichi (Hu et al., 2019), which we find to be faster than as reported in the original paper.

Given a $D \times D$ prior noise image $I_W \in \mathbb{R}^{D \times D\,2}$ and a deformation map $\psi : [0,1]^2 \to [0,1]^2$, the noise-warping algorithm (Chang et al., 2024) computes the warped noise image $\widetilde{I}_W \in \mathbb{R}^{D \times D}$ with upsampling level $N \in \mathbb{Z}_{\geq 1}$ as follows:

1. For $i, j = 1, \ldots, D$, upsample noise pixel $[I_W]_{i,j}$ to an $N \times N$ subimage $[\widehat{I}_W]_{i,j} \in \mathbb{R}^{N \times N}$:

$$[\widehat{I}_W]_{i,j} = \frac{[I_W]_{i,j}}{N^2} + \frac{1}{N}\left(Z - \frac{S}{N^2}\right), \text{ with } Z \sim \mathcal{N}(\mathbf{0}, \mathbf{I}) \text{ and } S = \sum_{k=1}^{N^2} Z_k. \quad (1)$$

The subimage for each pixel assembles into an $ND \times ND$ upsampled noise image $\widehat{I}_W$.

2. For $i, j = 1, \ldots, D$, the pixel square $A_{i,j} := [\frac{i-1}{D}, \frac{i}{D}] \times [\frac{j-1}{D}, \frac{j}{D}]$ is warped to a deformed pixel region $\widetilde{A}_{i,j} := \psi(A_{i,j})$, and the warped noise pixel $[\widetilde{I}_W]_{i,j}$ is set to be the sum of all subpixels in $\widehat{I}_W$ covered by $\widetilde{A}_{i,j}$ divided by $\sqrt{|\widetilde{A}_{i,j}|}$, where $|A|$ denotes the Lebesgue measure of a Borel set $A \subset \mathbb{R}^2$.

We describe an alternative but equivalent procedure by making the following two observations, which are illustrated in Figure 1.

**Gathering Noise $\to$ Scattering Noise.** While the original procedure computes the warped noise image by *gathering* the upsampled noise subpixels in each deformed pixel region $\widetilde{A}_{i,j}$ in a *Lagrangian* fashion, we can instead use an alternative procedure by *scattering* the upsampled noise subpixels in each pixel square $A_{i,j}$ to overlapping deformed pixel regions. This new *Eulerian* procedure does not change the output, but it yields new insights in conjunction with our second observation.

**Scattering Noise $\to$ Counting Overlapping Subpixels.** Observe that the $N \times N$ subpixels in $[\widehat{I}_W]_{i,j}$, for every $i, j$, are correlated only through their sum $S$ when conditioning on $[I_W]_{i,j}$ (1), so they are exchangeable. Hence, when scattering these upsampled noise subpixels to deformed pixel regions, the order of scattering does not matter, and we only need to count *the number of subpixels* covered by each deformed pixel region.

**Alternative Eulerian Procedure.** Putting both observations together, we now describe an alternative procedure to Chang et al. (2024) with unaltered output:

1. For each noise image pixel $[I_W]_{i,j}$, draw an upsampled subimage, now represented as a 1D vector $X \in \mathbb{R}^{N^2}$ using (1). Then, compute a prefix sum vector $H_{i,j}$ via $[H_{i,j}]_k := \sum_{q=1}^{k} X_q$ for $k = 1, \ldots, N^2$.

2. Warp each pixel square and compute deformed pixel regions $\widetilde{A}_{i,j}$ as before.

3. For each $A_{i,j}$, let $M$ denote the number of deformed pixel regions that overlap with $A_{i,j}$. With index $k = 1, \ldots, M$, we use $l_k, m_k$ to denote the coordinates of the $k^{\text{th}}$ overlap, whose pixel region is $\widetilde{A}_{\ell_k, m_k}$ and pixel value $[\widetilde{I}_W]_{\ell_k, m_k}$. Form $L \in \mathbb{Z}_{\geq 0}^M$ where $L_k$ represents the number of upsampled subpixels covered by $\widetilde{A}_{\ell_k, m_k}$. Then, compute a prefix sum $[C_{i,j}]_k := \sum_{q=1}^{k} L_q$. For $k = 1, \ldots, M$, accrue $[H_{i,j}]_{[C_{i,j}]_k} - [H_{i,j}]_{[C_{i,j}]_{k-1}}$ to $[\widetilde{I}_W]_{\ell_k, m_k}$.

4. Divide each warped noise pixel $[\widetilde{I}_W]_{i,j}$ by $\sqrt{|\widetilde{A}_{i,j}|}$.

**Discussion.** Compared to the original procedure by Chang et al. (2024), this alternative but equivalent algorithm highlights how the upsampled subpixels of $[I_W]_{i,j}$ are scattered to form the warped noise pixels. In particular, each warped noise pixel receives the sum of a segment of $X$, which is computed by taking the difference of two entries of $H_{i,j}$. Since $H_{i,j}$ represents summations of weakly correlated and exchangeable subpixels, once conditioned on $[I_W]_{i,j}$, *can we avoid explicitly instantiating every single subpixel*, but instead model the *sum* of these weakly correlated subpixels?

The key insight of this paper is that when the upsampling resolution $N \to \infty$, the scaling limit of the prefix sum $H_{i,j}$ (with proper interpolation and time scaling to a continuous function) is precisely the Brownian bridge (Durrett, 2019, Section 8.4) conditioned on $[I_W]_{i,j}$. Once this connection is established, it is easy to progressively sample increments of the Brownian bridge, resulting in a clean and efficient noise-warping algorithm that bypasses the need for upsampling in Chang et al. (2024).

---

[2]Here we assume that the noise image is square and has a single channel only to simplify notation. In practice, the noise image can have arbitrary aspect ratio and number of independent channels.

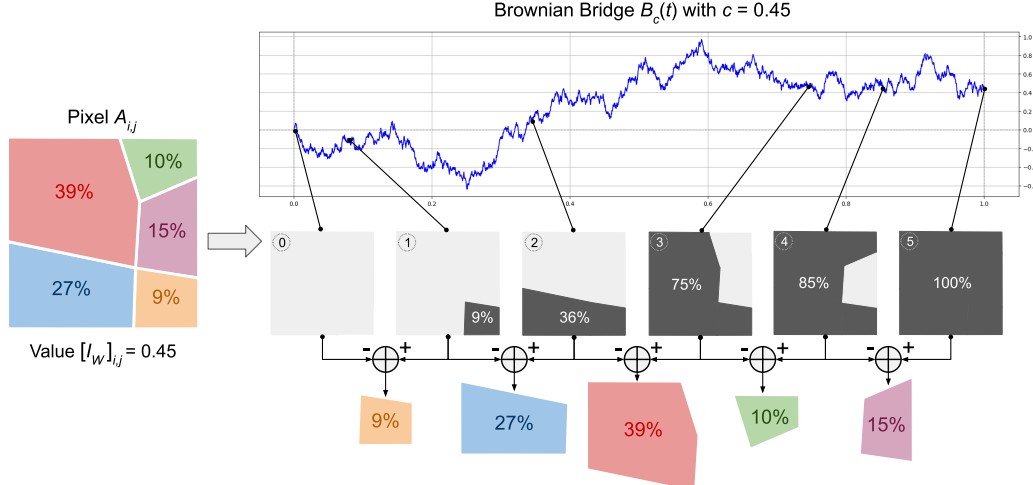

Figure 2: Connection between Eulerian noise warping (scattering) and increments of a Brownian bridge for a fixed prior noise pixel $[I_W]_{i,j}$. The overlapping area with each warped pixel region becomes the "time increments" — sampling the Brownian bridge at these "times" and taking consecutive differences yields the integral noise that is accrued to each warped noise pixel.

## 2.1 INFINITE-RESOLUTION NOISE SCATTERING

In this section, we first derive a scaling limit result to Brownian bridges. We then illustrate that the limiting version of the Eulerian procedure from the previous section matches precisely this scaling limit result. Lastly, we describe an autoregressive way to sample increments of a Brownian bridge that is linear in runtime in terms of the number of increments.

**Theorem 1** (Scaling limit to Brownian bridge). Let $\{Z_n\}$ be a sequence of i.i.d. random variables with finite variance that are normalized such that $\mathbb{E}[Z_n] = 0$ and $\text{Var}(Z_n) = 1$. For $c \in \mathbb{R}$, define

$$S_n := \sum_{i=1}^n Z_i, \qquad X_{i,n} := \frac{c}{n} + \frac{1}{\sqrt{n}} \left( Z_i - \frac{S_n}{n} \right).$$

Consider the sequence of random continuous functions $\{H_n(t)\} \subset C[0,1]$ defined as

$$H_n(t) := \sum_{i=1}^{\lfloor nt \rfloor} X_{i,n} + (nt - \lfloor nt \rfloor) X_{\lfloor nt \rfloor+1,n}.$$

Then the sequence $\{H_n\}$ converges in distribution under the sup-norm metric on $C[0,1]$ to $B_c(t) :=$ $W(t) - tW(1) + tc$, the Brownian bridge ending at $c$, where $W(t)$ is standard Brownian motion. Moreover, in distribution, we have $B_c(t) \stackrel{d}{=} (W(t) \mid W(1) = c)$, where $(W(t) \mid W(1) = c)$ is the disintegrated measure (Pachl, 1978) of $W(t)$ on $W(1) = c$.

We prove Theorem 1 in Appendix A. To connect the Eulerian procedure with the setup in Theorem 1, let us fix a pixel $[I_W]_{i,j}$, and let $B := B_{[I_W]_{i,j}}$, $H := H_{i,j}$, $C := C_{i,j}$ to simplify the notation. By setting $n = N^2$ and $c = [I_W]_{i,j}$, the sequence $\{X_{k,n}\}$ from the theorem has exactly the same law as the upsampled subpixels in $[\widehat{I}_W]_{i,j}$. Moreover, $H_{nt} = H_n(t)$ when $nt \in \mathbb{Z}_{\geq 1}$. By taking $N \to \infty$, implying $n \to \infty$, for any $t_1, \ldots, t_M \in [0,1]$, we have the convergence in distribution of $(H_{\lfloor nt_1 \rfloor}, \ldots, H_{\lfloor nt_M \rfloor}) \stackrel{d}{\to} (B(t_1), \ldots, B(t_M))$. Recall in the Eulerian procedure, we only need to access the prefix sum $H$ at indices $\{C_k\}_{k=1}^M$, where $C_k$ counts the number of upsampled subpixels covered by the first $k$ overlaps. This suggests that if we choose

$$t_k = \lim_{N \to \infty} \frac{C_k}{N^2} = \sum_{k'=1}^k \left| A_{i,j} \cap \widetilde{A}_{\ell_{k'}, m_{k'}} \right|,$$

and use $B(t_k)$ in place of $H_k$, then we just need to sample from $B$ at times $t_1, \ldots, t_M$ — precisely the limiting algorithm of the Eulerian procedure. We illustrate this connection in Figure 2.

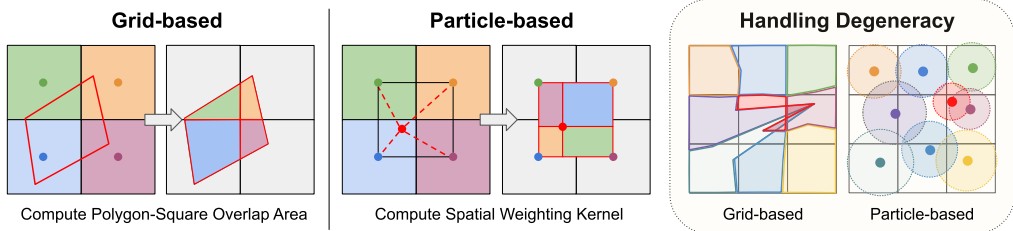

Figure 3: The grid-based variant (left) computes the overlapping areas by explicitly constructing the polygon for the deformed pixel region. The particle-based variant (middle) approximates these areas with a weighting kernel. With degenerate maps (right), the fixed topology of the grid-based variant can lead to problems, while the connectivity-free, particle-based variant remains stable.

**Autoregressive Sampling of Brownian Bridges.** Since a Brownian bridge is a Markov process (Oksendal, 2013, Exercise 5.11), we can sample the vector $(B_c(t_1), \ldots, B_c(t_M))$ in an autoregressive fashion, each time sampling $B_c(t_{k+1})$ conditioned on $B_c(t_k)$:

$$(B_c(t_{k+1}) \mid B_c(t_k) = q) \overset{d}{=} \mathcal{N}\left(\frac{1 - t_{k+1}}{1 - t_k}q + \frac{t_{k+1} - t_k}{1 - t_k}c, \frac{(t_{k+1} - t_k)(1 - t_{k+1})}{1 - t_k}\right). \quad (2)$$

Once the Brownian bridge at times $t_k$ is sampled, we just need to accrue the increments $B_c(t_k) - B_c(t_{k-1})$ to $[\widetilde{I}_W]_{\ell_k, m_k}$, the $k^{\text{th}}$ overlapped warped noise pixel. This allows us to present Algorithm 1. Compared to the discrete procedures described earlier, we no longer need upsampling. In addition, we exploited the autoregressive nature of Brownian bridges to bring down the time complexity to linear in the number of overlapping warped pixel regions.

---

**Algorithm 1** Infinite-Resolution Integral Noise Warp

---

**Input:** prior noise image $I_W \in \mathbb{R}^{D \times D}$, deformation map $\psi : [0, 1] \to [0, 1]$
**Output:** warped noise image $\widetilde{I}_W \in \mathbb{R}^{D \times D}$
  Build a partition record $\mathcal{P}$ from $\psi$ (Section 2.2)
  Initialize $\mathcal{A}_{i,j} \leftarrow 0$ for all $i, j = 1, \ldots, D$               $\triangleright$ $\mathcal{A}_{i,j}$ will contain *the area of* $\widetilde{A}_{i,j}$
  **parallel for each** $u, v = 1, \ldots, D$ **do**
    $t, q, M \leftarrow 0, 0, |\mathcal{P}_{u,v}|$
    **for** $k = 1, \ldots, M$ **do**
      $(a, i, j) \leftarrow [\mathcal{P}_{u,v}]_k$           $\triangleright$ $a$ is the overlapping area between $A_{i,j}$ and $\widetilde{A}_{u,v}$
      Sample $q' \sim (B_c(t + a)|B_c(t) = q)$ by (2) with $c = [I_W]_{u,v}$
      $[\widetilde{I}_W]_{i,j} \leftarrow [\widetilde{I}_W]_{i,j} + (q' - q)$
      $\mathcal{A}_{i,j} \leftarrow \mathcal{A}_{i,j} + a$
      $q, t \leftarrow q', t + a$
  Normalize $[\widetilde{I}_W]_{i,j} \leftarrow \mathcal{A}_{i,j}^{-\frac{1}{2}}[\widetilde{I}_W]_{i,j}$ for all $i, j = 1, \ldots, D$
  **return** $\widetilde{I}_W$

---

**Preservation of Gaussian White Noise.** A central desideratum of noise warping is that the resulting warped noise image $\widetilde{I}_W$ needs to have pixels that are i.i.d. standard Gaussians when the prior noise image $I_W$ is Gaussian white noise. This ensures that the warped noise is in-distribution for a pre-trained diffusion model. Our algorithm automatically guarantees this preservation of Gaussianity, as long as the warping function $\psi$ is injective. To see this, the injectivity of $\psi$ implies that the warped pixel regions are non-overlapping in the square $[0, 1]^2$. For each $A_{i,j}$, since $[I_W]_{i,j} \overset{d}{=} \mathcal{N}(0, 1) \overset{d}{=} W(1)$, by the conditional interpretation of Brownian bridges (1), when marginalizing out $[I_W]_{i,j}$, the Brownian bridge $B_{[I_W]_{i,j}}$ reduces to standard Brownian motion. Since the increments of the Brownian motion are independent Gaussians, the contribution to a deformed pixel region is simply a zero-mean Gaussian with variance equal to the overlapping area. Therefore, each deformed pixel region will receive the sum of a number of independent Gaussians whose variances sum to the area of the region. The scaling by the inverse square root of the area in Algorithm 1 thus makes each warped noise pixel an i.i.d. standard Gaussian.

## 2.2 Building Partition Records

To compute Algorithm 1, we need a method to compute the partition record $\mathcal{P}$ which specifies how each pixel square is partitioned by multiple deformed pixel regions. In this section, we present one grid-based (Algorithm 2) and one particle-based (Algorithm 3) method for building $\mathcal{P}$. In particular, for each pixel square with indices $(u, v)$, we compute $\mathcal{P}_{u,v}$ as a list of 3-tuples $(a, i, j)$, where $(i, j)$ identifies the overlapped deformed pixel region and $a$ represents the overlapping area.

| **Algorithm 2** Grid-based Partition | **Algorithm 3** Particle-based Partition |
|---|---|
| **Input:** Deformation map $\psi$ | **Input:** Deformation map $\psi$ |
| **Output:** Partition record $\mathcal{P}$ | **Output:** Partition record $\mathcal{P}$ |
| 1: **parallel for each** $i, j$ **do** | 1: **parallel for each** $i, j$ **do** |
| 2: $\quad A^* \leftarrow \text{DiscretizeSquare}(A_{i,j})$ | 2: $\quad (x, y) \leftarrow \psi(\frac{i+0.5}{D}, \frac{j+0.5}{D})$ |
| 3: $\quad S \leftarrow \psi(A^*)$ | 3: $\quad \alpha_{0,0}, \alpha_{0,1}, \alpha_{1,0}, \alpha_{1,1} \leftarrow \text{BilinearWeights}(X)$ |
| 4: $\quad u^-, u^+, v^-, v^+ \leftarrow \text{AABB}(S)$ | 4: $\quad$ **for** $s, t \in [0, 1]$ **do** |
| 5: $\quad$ **for** $u \in [u^-, u^+]$ **do** | 5: $\quad\quad u, v \leftarrow \lfloor x \rfloor + s, \lfloor y \rfloor + t$ |
| 6: $\quad\quad$ **for** $v \in [v^-, v^+]$ **do** | 6: $\quad\quad \mathcal{P}_{u,v} \leftarrow \mathcal{P}_{u,v} + [(\alpha_{s,t}, i, j)]$ |
| 7: $\quad\quad\quad a \leftarrow \text{PolygonArea}(\text{Clip}(S, u, v))$ | 7: **parallel for each** $u, v$ **do** |
| 8: $\quad\quad\quad \mathcal{P}_{u,v} \leftarrow \mathcal{P}_{u,v} + [(a, i, j)]$ | 8: $\quad$ Normalize total area of $\mathcal{P}_{u,v}$ to $D^{-2}$ |
| 9: **return** $\mathcal{P}$ | 9: **return** $\mathcal{P}$ |

As illustrated in Figure 3, our grid-based method (left) follows Chang et al. (2024) by modeling each deformed pixel region as an octagon and computes overlapping areas by clipping it against undeformed pixel squares; our particle-based method (middle) borrows from the grid-to-particle techniques in fluid particle-in-cell methods (Brackbill et al., 1988), where we treat each deformed pixel region as a particle and each undeformed pixel square as a grid cell. Each particle requests area from nearby cells based on distance; upon receiving requests, each cell normalizes the requests to ensure partition-of-unity, and distributes its area to contacting particles.

**Discussion.** Conceptually, our grid and particle-based methods correspond to two different interpretations of $\psi$ when provided as discrete samples (*e.g.*, an optical flow image). The grid-based method implicitly reconstructs the continuous $\psi$ field by linear interpolation, whereas the particle-based method assumes $\psi$ is only known point-wise. The implication is that when $\psi$ is smooth, linear interpolation works well and the grid-based method will yield a higher-quality warp as seen in Figure B.3. But when $\psi$ is non-smooth, which is commonly the case in real world, linear interpolation can lead to degenerate polygons as illustrated on the right of Figure 3. The spurious overlaps between the degenerate polygons will lead to spatial correlation in the warped noise image. Although both Chang et al. (2024) and our grid-based method implement fail-safes[3] to avoid noise sharing and maintain spatial independence in practice, they suffer from the intrinsic ambiguity caused by these overlaps. On the other hand, the particle-based method circumvents such overlaps to begin with.

In addition, we highlight the simplicity and parallelizability of the particle-based method, as it boils down the computation of $\mathcal{P}$ to evaluating one bilinear kernel per pixel. Leveraging this fact, we can very conveniently and efficiently extend our noise warping algorithm to higher spatial dimensions by replacing the bilinear kernel to its higher-dimensional counterparts, as shown in Figure 8.

## 3 Results

We verify our theoretical claims by showing that both variants of our method preserve Gaussian white noise distribution, and that Chang et al. (2024) (HIWYN) converges to our grid-based variant as $N$ increases. We analyze the behaviors of our grid and particle-based variants under diffeomorphic and non-diffeomorphic deformations. We then apply our method to video generation tasks and benchmark against existing methods (Ge et al., 2023; Chen et al., 2023; Chang et al., 2024). Finally, we extend our method to warping volumetric noise and demonstrate a use case in 3D graphics.

**Gaussian White Noise Preservation.** In Figure 4, we iteratively warp a noise image by the same deformation map for 50 timesteps. We gauge the output noise's resemblance to Gaussian white noise through 1) measuring normality using one-sample Kolmogorov-Smirnov (K-S) test, and 2) detecting

---

[3]In our case, we clamp the input $t_{k+1}$ to 1 before sampling by 2. Intuitively, this means that when an undeformed pixel has assigned its entire pixel region to deformed pixels, later noise requests will be neglected.

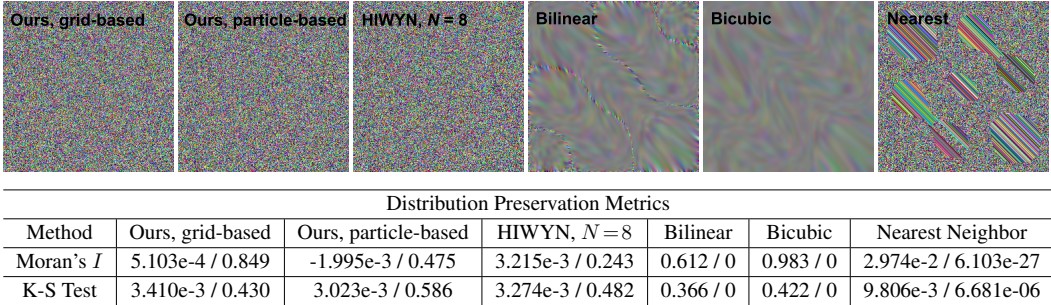

| | Distribution Preservation Metrics | | | | | |
|---|---|---|---|---|---|---|
| Method | Ours, grid-based | Ours, particle-based | HIWYN, $N=8$ | Bilinear | Bicubic | Nearest Neighbor |
| Moran's $I$ | 5.103e-4 / 0.849 | -1.995e-3 / 0.475 | 3.215e-3 / 0.243 | 0.612 / 0 | 0.983 / 0 | 2.974e-2 / 6.103e-27 |
| K-S Test | 3.410e-3 / 0.430 | 3.023e-3 / 0.586 | 3.274e-3 / 0.482 | 0.366 / 0 | 0.422 / 0 | 9.806e-3 / 6.681e-06 |

Figure 4: Preservation of Gaussian white noise achieved by different warping methods. We report scores and p-values for both Moran's $I$ (spatial correlation) and K-S test (normality). Under these metrics, the results from our method (both variants) and HIWYN are indistinguishable from white Gaussian noise, while generic, interpolation-based warping methods lead to corrupted noise.

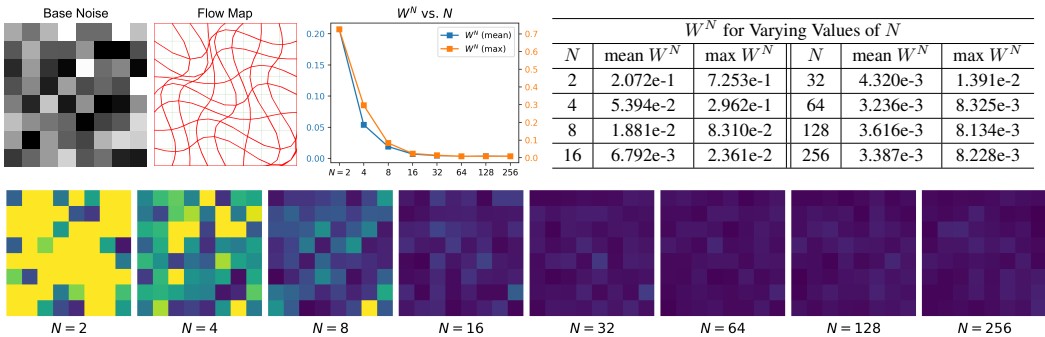

| | $W^N$ for Varying Values of $N$ | | | | |
|---|---|---|---|---|---|
| $N$ | mean $W^N$ | max $W^N$ | $N$ | mean $W^N$ | max $W^N$ |
| 2 | 2.072e-1 | 7.253e-1 | 32 | 4.320e-3 | 1.391e-2 |
| 4 | 5.394e-2 | 2.962e-1 | 64 | 3.236e-3 | 8.325e-3 |
| 8 | 1.881e-2 | 8.310e-2 | 128 | 3.616e-3 | 8.134e-3 |
| 16 | 6.792e-3 | 2.361e-2 | 256 | 3.387e-3 | 8.228e-3 |

Figure 5: Convergence of HIWYN to our method as $N$ increases. Top left: experimental setup with prior noise and deformation map. Top middle: 2-Wasserstein distance $W^N$ between the output of HIWYN and ours. Top right: statistics table. Bottom: $W^N$ difference image between the output of HIWYN and ours as $N$ increases. $W^N$ becomes statistically insignificant for $N \geq 64$.

spatial correlation using Moran's $I$. Our results show that both HIWYN and our method generate warped noise images that are indistinguishable from Gaussian white noise under these metrics, while interpolation-based baselines deviate significantly from the desired distribution.

**Convergence of Chang et al. (2024).** We validate that our method is the limiting case of HIWYN. Starting with an $8 \times 8$ prior noise image and a flow map (Figure 5, top left), we run our method along with HIWYN for $N \in \{2, 4, 8, \dots, 256\}$ for 100,000 independent runs to estimate the distribution of the warped noise image. For each upsampling resolution $N$, we compute the 2-Wasserstein distance $W^N$ between the output of HIWYN and that of our method. The results in Figure 5 demonstrate the convergence of HIWYN to our method as $N$ increase, and reveal that $N=8$ (recommended by Chang et al. (2024)) is not yet in the converged phase to yield a negligible $W^N$.

**Performance Comparison.** For our methods and HIWYN with upsampling levels $N \in \{2, 4, 8\}$, we perform 100 independent runs on a $1024 \times 1024$ image. We report the kernel time with CPU and GPU backends (Figure 6) as well as the memory usage. The runtime and memory usage of our methods are largely comparable with those of HIWYN with $N = 2$. Compared to HIWYN with $N = 8$, both variants of our method offer order-of-magnitude improvements in runtime and memory usage. Specifically, our grid-based method achieves infinite upsampling resolution while being $19.7\times$ faster on CPU and $8.0\times$ faster on GPU, using $9.22\times$ less memory, and our particle-based method, albeit not strictly equivalent to HIWYN at $N = \infty$, achieves a $41.7\times$ speedup on GPU. In the following sections, we show that the particle-based version consistently achieves comparable quality to the grid-based version in real-world scenarios.

**Comparison between Grid-Based and Particle-Based Variants.** In Figure B.3, we compare both variants when the deformation map is diffeomorphic under different levels of distortion. Visually, the difference between the two variants is negligible at frame 25 and becomes noticeable in frame

| Comparison of Time and Memory Costs | | | |
|---|---|---|---|
| Method | Time (CPU) | Time (GPU) | Memory |
| $N = 2$ (HIWYN) | 19.30s | 2.597s | 293.7MB |
| $N = 4$ (HIWYN) | 75.33s | 9.247s | 746.6MB |
| $N = 8$ (HIWYN) | 398.3s | 35.91s | 2147MB |
| $N = \infty$ (ours, grid) | 20.16s | 4.491s | 232.9MB |
| $N = \infty$ (ours, particle) | 15.68s | 0.862s | 320.9MB |

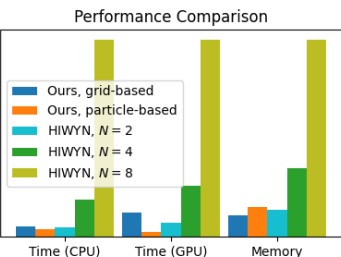

Figure 6: Runtime and memory usage of our method versus HIWYN with $N = 2, 4, 8$. We compare total allocated memory and kernel time on a CPU/GPU. The computation is done on a laptop with Intel i7-12700H and Nvidia RTX 3070 Ti.

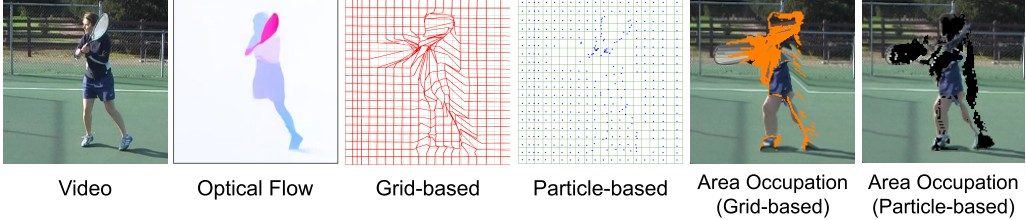

Figure 7: Comparison of grid-based versus particle-based variants under non-diffeomorphic optical flow. Pixels with area contention are colored in orange. Further results are given in Figure B.5.

100. We measure this difference by comparing the deformed regions for each pixel in terms of IoU and weighted Chamfer distance. We additionally compare the particle-based result with that of an identity-map baseline (right column in Figure B.3), which shows that the gap between the two variants remains small even under large distortion. In Figure 7, we stress test both variants under non-diffeomorphic maps obtained using optical flow (Teed & Deng, 2020) on a real-world video (Brox & Malik, 2011). In panels 3 and 4, we see that the real-world flow map induces inverted meshes for the grid-based variant and clustered particles for the particle-based variant. While clustered particles are always assigned disjoint regions due to the continuous nature of our algorithm, mesh inversions cause area contention issues (non-injectivity) due to overlaps. In panels 5 and 6, we mark the grid cells with area contention in orange, which occurs in the grid-based version but not in the particle-based version.

**Video Super-resolution with I²SB.** We integrate our method with I²SB (Liu et al., 2023) and adapt its pre-trained *image* $4\times$ super-resolution model (bicubic) to perform *video* super-resolution. We show our results in Figures 9 and B.1, and refer to our supplementary video for better visualization of these results. Since I²SB is an image-to-image bridge model, it well preserves the low-frequency structures of the input images regardless of noise scheme. But as seen in our video, without noise warping, the results either show strong flickering in the high-frequency details (random noise) or sticking artifacts (fixed noise). Noise warping allows high-frequency details to transport with the optical flow, making the result significantly more consistent. We also validate that both our variants yield visual quality on par with HIWYN across all tested scenarios while being much more efficient.

**Conditional Video Generation with SDEdit.** We apply our method to conditional video generation by adapting SDEdit (Meng et al., 2021), a conditional image generation method, to producing consistent video frames. We set SDEdit's parameter $t_0$ to 0.4, and apply Perturbed-Attention Guidance (Ahn et al., 2024) with a strength of $3.0$. Our two inputs are a conditioning video (generated by applying a median filter to real-world videos similar to (Chen et al., 2023)) and an optical flow field (Teed & Deng, 2020). Without noise manipulation, if we run SDEdit independently frame-by-frame (Figure B.6, *random*), the high-frequency details display significant flickering. By warping the noise using the optical flow, the temporal consistency is much improved, and we observe that our method (both variants) and HIWYN yield comparable visual qualities. Full experimental results with comparisons to Control-A-Video (Chen et al., 2023), PYoCo (Ge et al., 2023) and additional baselines are provided in Figures B.6 and B.7 with quantitative statistics reported in Table 1. Further results that additionally integrate cross-frame attention (Ceylan et al., 2023) (anchored every 3 frames) are shown in Figure 10 and B.2. We refer to our supplementary video for better visualization.

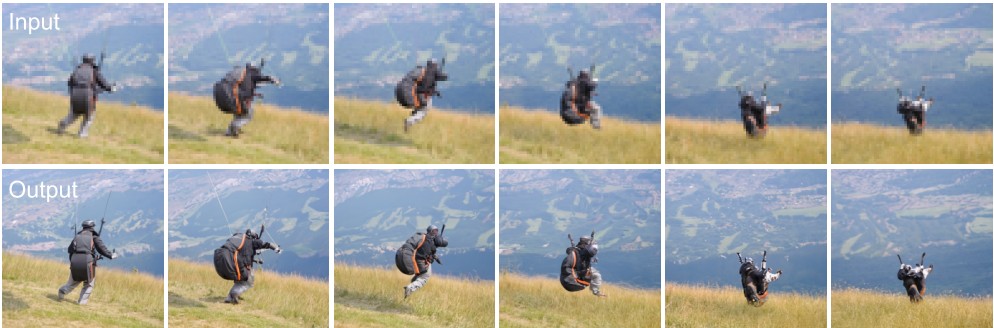

Figure 9: Video $4\times$ super-resolution by integrating our method (particle) with I²SB. Top row shows the low resolution input video; bottom row shows the output video. Additional results are shown in Figure B.1. We refer to our supplementary video for better visualization of these results.

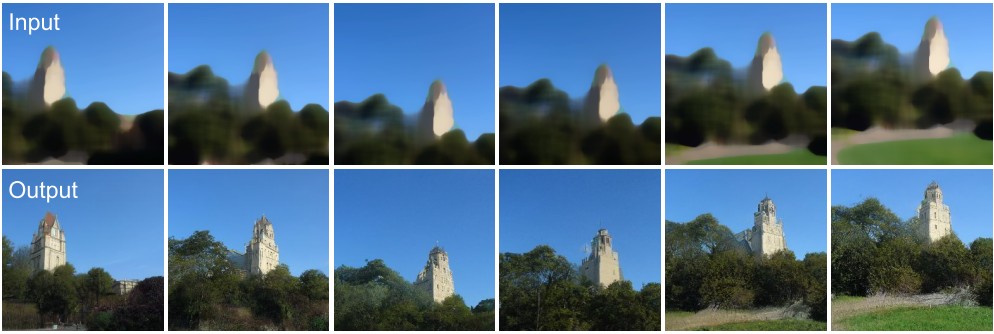

Figure 10: Conditional video generation results by integrating our method (particle) with SDEdit and cross-frame attention (Ceylan et al., 2023). Top row shows the input video prompt in a stroke painted style which is converted into a video of photorealistic style (bottom). Additional results are shown in Figure B.2. We refer to our supplementary video for better visualization of these results.

**3D Noise Warp.** We extend our particle-based algorithm to 3D by replacing the bilinear kernel with the trilinear kernel in Algorithm 3 and apply it to GaussianCube (Zhang et al., 2024), which denoises a dense 3D noise grid to reconstruct 3D Gaussians. We adapt it to perform conditional generation a la SDEdit. Starting with a 3D pickup truck generated unconditionally, we condition the model to generate vehicles with smaller and larger cabins by deforming the truck with a horizontal shear velocity field. We compare the results obtained by using random noise to those using noise warped with our particle-based method. We show the results in Figure B.4 and refer to our supplementary video for better visualization.

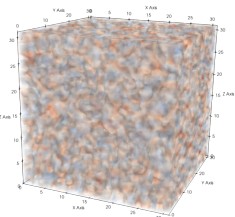

Figure 8: 3D noise warped by our method.

## 4 RELATED WORKS

**Noise in Diffusion Models.** Diffusion models generate images from input noise, and noise can thus be considered the counterpart to the latent codes utilized in GAN models. As such, the outputs of diffusion models have dependencies and correlations to the initial input noise, making noise a useful handle to control temporal consistency (Khachatryan et al., 2023). In addition to Chang et al. (2024) which this work was inspired by and improves upon, there are various other temporal noise manipulation techniques that do not preserve Gaussian noise distribution– some methods (Ma et al. (2024); Ren et al. (2024)) blend high frequency Gaussian noise with low frequency motion, while others (Mokady et al. (2022); Wallace et al. (2022)) rely on approximating the inversion of noise from temporally coherent image sequences. Pandey et al. (2024) goes one step further and manipulates inverted noise in 3D space. These approaches are flexible but degrade the output of the diffusion model due to the domain gap between inference time noise and training time noise, and as such, have occasionally been accompanied by mitigation strategies such as anisotropic diffusion (Yu et al. (2024)). Noise manipulation is also not limited to the generation and stylization of videos,

but has various applications in image editing (Hou et al. (2024); Pandey et al. (2024)) and 3D mesh texturing (Richardson et al. (2023)) as well.

**Noise in Computer Graphics.** While our noise warping work draws main inspiration from simulation techniques, spatial noise manipulation has been extensively studied in the graphics community through applications in animation and rendering. Works like (Kass & Pesare, 2011; Burley et al., 2024) present 2D noise manipulation techniques that add a stylized organic hand-drawn look to computer-generated animation via dynamic noise textures (Perlin, 1985). In order to make sure the stylization is temporally consistent and visually pleasing, noise textures are deformed in a way that makes them consistent with the underlying animation, but little emphasis is given to the preservation/rigor of the noise distribution. On the other hand, properties of 2D spatial noise have been extensively and rigorously studied in rasterization and raytracing literature (Cook, 1986; Lagae & Dutré, 2008), originating from the idea of using dithering to reduce banding and quantization artifacts in image signal processing (Roberts, 1962). In particular, the lack of low frequency details and clumping in blue noise as opposed to white Gaussian noise has made it the choice of foundational antialiasing methods such as Poisson disc sampling (McCool & Fiume, 1992), and recent progress made in this line of antialiasing research has close ties with our methodology. For example, Wolfe et al. (2022) look at accelerating rendering tasks by extending spatial blue noise to the temporal domain, while Huang et al. (2024) show promising results in supplementing white noise with blue noise during diffusion model training.

## 5   DISCUSSION AND CONCLUSION

We presented *infinite-resolution integral noise warping*, a novel algorithm for computing temporally coherent, distribution-preserving noise transport to guide diffusion models to produce consistent results. By deriving a continuous-space analogy to the existing upsampling-based strategy (Chang et al., 2024), our method not only further improves the accuracy by effectively raising the upsampling resolution to infinity, but also drastically reduces the computational cost.

**Usability of Noise Warping**   We highlight that the noise warping problem that we address is a recurring subtask in generative modeling, and our method is hence a general-purpose tool that can be integrated in a variety of ways that extend well beyond the ones we showcase in the paper. First, noise warping, which excels at controlling high-frequency details, is orthogonal and thus combinable with *feature-level*, structure-preserving techniques (*e.g.* Ceylan et al. (2023); Cong et al. (2023)) to achieve consistency across the frequency spectrum. Our drastic cost-saving makes noise warping an affordable and harm-free add-on to all such existing and future techniques. In addition, the concurrent work by Daras et al. (2024) shows that noise warping can be combined with equivariance guidance to gain further consistency and integrate with latent diffusion models like SDXL (Podell et al., 2023). Beyond video generation, Kwak et al. (2024) showcases the usefulness of noise warping in 3D generation by combining with score distillation sampling (SDS). The advanced noise warping algorithm that we propose presents itself as a desirable candidate across these diverse tasks.

**Significance of Our Speed-up**   We argue that the drastic speed-up our method offers has profound practical significance. While the standard denoising diffusion setting requires only a single noise warp operation per image, there exist many use cases that require noise warping to be computed more intensively, which renders our speed-up critical. For example, the combination with bridge models (*e.g.* I$^2$SB) requires one noise warp per iteration. With its reported $> 0.6$s time cost per warp, preparing the noise using HIWYN would cost $\sim 4\times$ the time to actually run the image generation model, increasing the total inference time from $\sim 24$ minutes to $\sim 2$ hours. In comparison, our method (particle) prepares the noise in $40.6$s (wall time), effectively making the overhead negligible. Similarly, combining noise warping with SDS also requires one noise warp per iteration, which makes HIWYN computationally intractable (Kwak et al., 2024) and our improvements called for. Our speed-up hence makes integral noise warping deployable in a much broader class of problems.

We identify a few directions for future work. First, our particle-based variant currently does not capture area contraction or expansion which may be addressed in the future with Voronoi partitioning. Secondly, our method can be readily combined with other map types such as UV maps to facilitate multi-view consistency. Furthermore, since our method relies on the consistency and quality of the deformation map, it can benefit from future advancements in flow extraction techniques. Finally, the effectiveness of warped volumetric noise in 3D generation and editing remains to be studied.

ACKNOWLEDGEMENT

We thank Lukas Lepicovsky, Ioan Boieriu, David Michielsen, Mohsen Mousavi, and Perry Kain from Eyeline Studios, for providing data that kickstarted this project and for assisting in shaping our research with practical future use cases. We also thank Austin Slakey for sharing his insights on training diffusion models, and Tianyi Xie for filming his cat for our testing.

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

## A  PROOF OF THEOREM 1

*Proof.* By unrolling the definitions, for $t \in [0, 1]$, we have

$$H_n(t) = S_n^*(t) - tS_n^*(1) + tc, \qquad S_n^*(t) \coloneqq \frac{1}{\sqrt{n}} \left( \sum_{i=1}^{\lfloor nt \rfloor} Z_i + (nt - \lfloor nt \rfloor) Z_{\lfloor nt \rfloor + 1} \right).$$

By Mörters & Peres (2010, Theorem 5.22), $\{S_n^*\}_{n \in \mathbb{Z}_{\geq 1}}$ converges in distribution to $W(t)$ under the sup-norm metric of $C[0, 1]$. To lift this convergence to the sequence $\{H_n\}_{n \in \mathbb{Z}_{\geq 1}}$, observe that the function $g : C[0, 1] \to C[0, 1]$ defined by

$$g(x(t)) \coloneqq x(t) - tx(1) + tc$$

is continuous under the sup-norm metric. To verify this, suppose $\lim_{n \to \infty} f_n = f$ for $\{f_n\}_{n \in \mathbb{Z}_{\geq 1}}, f \in C[0, 1]$. Then

$$\|g(f_n) - g(f)\|_\infty = \sup_{t \in [0,1]} |(f_n(t) - tf_n(1) + tc) - (f(t) - tf(1) + tc)|$$

$$\leq \|f_n - f\|_\infty + \|f_n(1) - f(1)\| \leq 2\|f_n - f\|_\infty \to 0.$$

Hence, by the continuous mapping theorem,

$$g(S_n^*) = H_n \xrightarrow{d} B(t) - tB(1) + tc.$$

To show

$$W(t) - tW(1) + tc = (W(t) \mid W(1) = c),$$

first of all, the conditioning $(W(t) \mid W(1) = c)$ is interpreted as the limit of $(W(t) \mid |W(1) - c| < \epsilon)$ as $\epsilon \to 0$. Denote $Y(t) \coloneqq W(t) - tW(1)$, so that $W(t) = Y(t) + tW(1)$. Since $\mathrm{Cov}(Y(t), tW(1)) = \mathrm{Cov}(W(t) - tW(1), tW(1)) = t\mathrm{Cov}(W(t), W(1)) - t^2\mathrm{Var}(W(1), W(1)) = 0$ and that $Y(t), tW(1)$ are jointly Gaussian, they are independent. Therefore,

$$\lim_{\epsilon \to 0}(W(t) \mid |W(1) - c| < \epsilon) = \lim_{\epsilon \to 0}(Y(t) + tW(1) \mid |W(1) - c| < \epsilon)$$

$$= Y(t) + \lim_{\epsilon \to 0}(tW(1) \mid |W(1) - c| < \epsilon)$$

$$= W(t) - tW(1) + tc.$$

$\square$

## B  ADDITIONAL RESULTS

In this section we include additional visual and numerical results. Figure B.1 shows additional video $4\times$ super-resolution results with I²SB in addition to those in Figure 9. Figure B.2 shows additional video generation results with SDEdit and cross-frame attention, which mirrors the setup of Figure 10. Figure B.6 and Figure B.7 show video generation results with SDEdit only (without cross-frame attention) to isolate the influence of noise initialization, and we compare our method (both variants) with HIWYN (Chang et al., 2024), Control-A-Video (Chen et al., 2023) PYoCo (Ge et al., 2023), as well as baselines with random, fixed, and interpolated noise using bilinear and nearest interpolating schemes. The statistics for both the church and cat scenes are reported in Table 1. Figure B.3 demonstrates the differences between both our variants in computing warped pixel regions in the diffeomorphic case. Figure B.5 shows the differences in the non-diffeomorphic case, where we use additional examples to demonstrate the area contention issue caused by degenerate meshes that applies similarly to our grid-based variant and (Chang et al., 2024), and highlight the robustness of our particle-based variant. Figure B.4 shows additional results for applying our particle-based, volumetric noise warping to 3D tasks with GaussianCube (Zhang et al., 2024).

| Video Generation Quality (Church) | | | | | | | | |
|---|---|---|---|---|---|---|---|---|
| Metric | Ours (G) | Ours (P) | HIWYN | PYoCo | CaV | Random | Fixed | Bilinear | Nearest |
| *Consistency* ↓ | 6.955e-2 | 7.057e-2 | 6.953e-2 | 9.800e-2 | 1.108e-1 | 1.150e-1 | 9.273e-2 | 6.906e-2 | 8.985e-2 |
| *Realism* ↓ | 7.251e-2 | 7.130e-2 | 7.197e-2 | 7.535e-2 | 7.430e-2 | 6.336e-2 | 5.893e-2 | 2.102e-1 | 8.194e-2 |
| *Faithfulness* ↓ | 1.111e-2 | 1.166e-2 | 1.171e-2 | 1.184e-2 | 1.555e-2 | 1.344e-2 | 1.030e-2 | 1.883e-2 | 3.090e-2 |
| Video Generation Quality (Cat) | | | | | | | | |
| Metric | Ours (G) | Ours (P) | HIWYN | PYoCo | CaV | Random | Fixed | Bilinear | Nearest |
| *Consistency* ↓ | 4.016e-2 | 4.070e-2 | 3.810e-2 | 5.995e-2 | 4.507e-2 | 1.093e-1 | 4.537e-2 | 2.613e-2 | 7.974e-2 |
| *Realism* ↓ | 2.235e-1 | 2.126e-1 | 2.108e-1 | 1.647e-1 | 1.931e-1 | 1.561e-1 | 1.982e-1 | 3.271e-1 | 3.195e-1 |
| *Faithfulness* ↓ | 7.841e-3 | 7.556e-3 | 7.305e-3 | 9.290e-3 | 7.427e-3 | 9.637e-3 | 8.558e-3 | 1.162e-2 | 1.128e-1 |

Table 1: We show the quantitative statistics for conditional video generation using SDEdit without cross-frame attention. The consistency is measured using Warp MSE (Chang et al., 2024), and the realism and faithfulness are measured as in Meng et al. (2021).

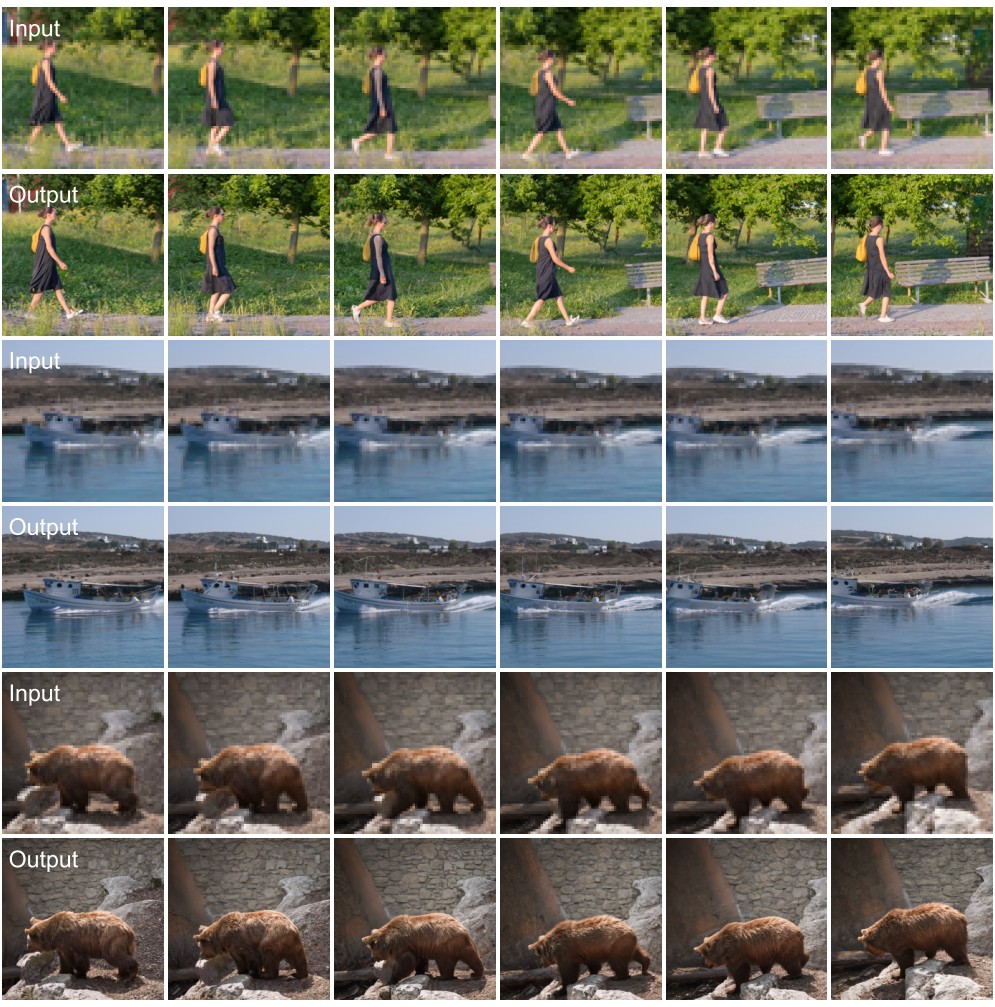

Figure B.1: Additional results generated by performing $4\times$ video super-resolution with $I^2$SB. For each scene, the upper row represents the low resolution input video, and the lower row represents the high resolution output video generated with our particle variant. We refer to our supplementary video for better visualization of these results.

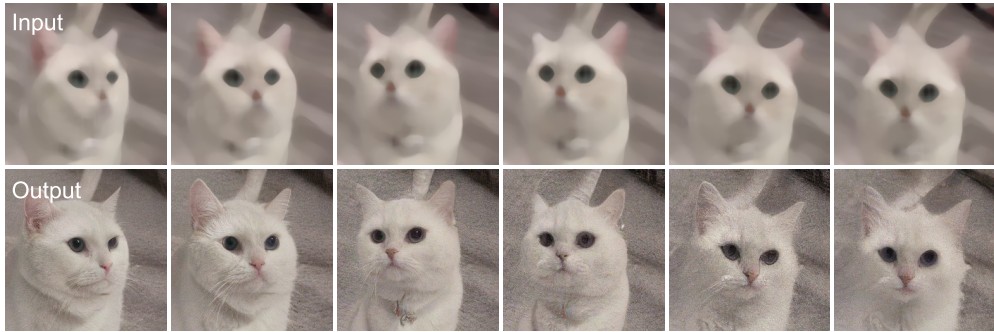

Figure B.2: Additional conditional video generation results by integrating our method (particle) with SDEdit and cross-frame attention (Ceylan et al., 2023) on a cat scene.

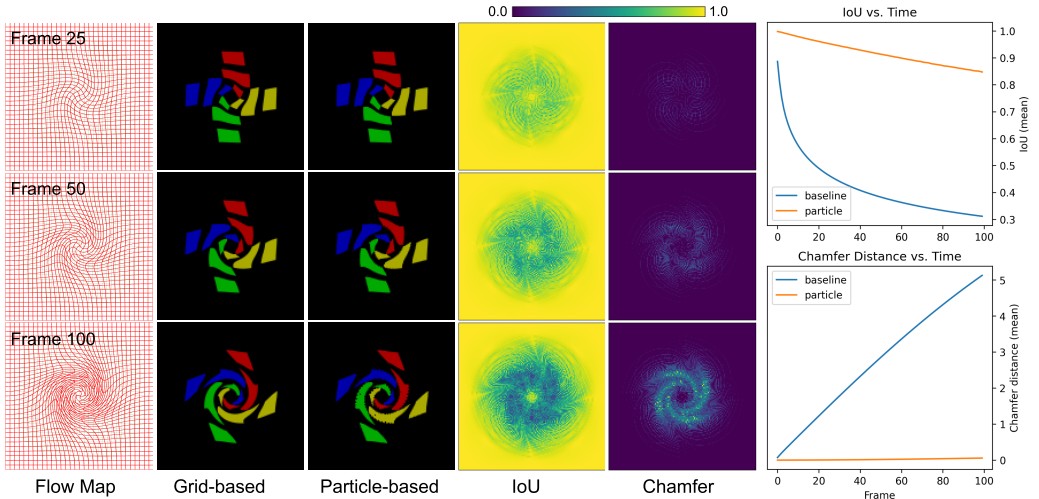

Figure B.3: Comparison between the grid-based and particle-based variants for building partition records when the deformation map is diffeomorphic. The first column shows the deformation map at different frames. The second and third columns visualize warped pixel regions for both methods. The fourth and fifth columns show the IoU (larger is better) and Chamfer distance (smaller is better) between the warped pixel regions of both variants. We plot the distance between particle and grid variants alongside a baseline, which is the distance between identity map and the grid variant, to verify that the particle-based version remains close to the grid-based version even under large distortion.

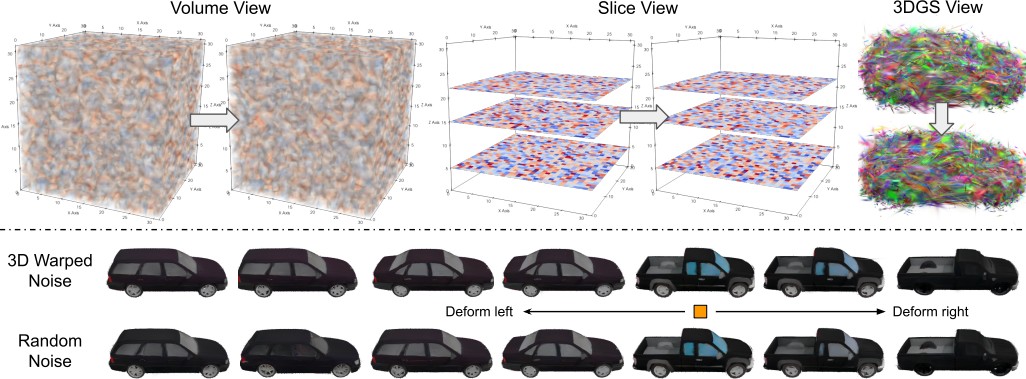

Figure B.4: Extension of our particle-based method to warping 3D, volumetric noise. We show volume renders of the noise on the top left, slice views on the top middle, and 3D Gaussians as used in the GaussianCube representation on the top right. The bottom row shows the respective results obtained by running 3D generation with warped and random 3D noise.

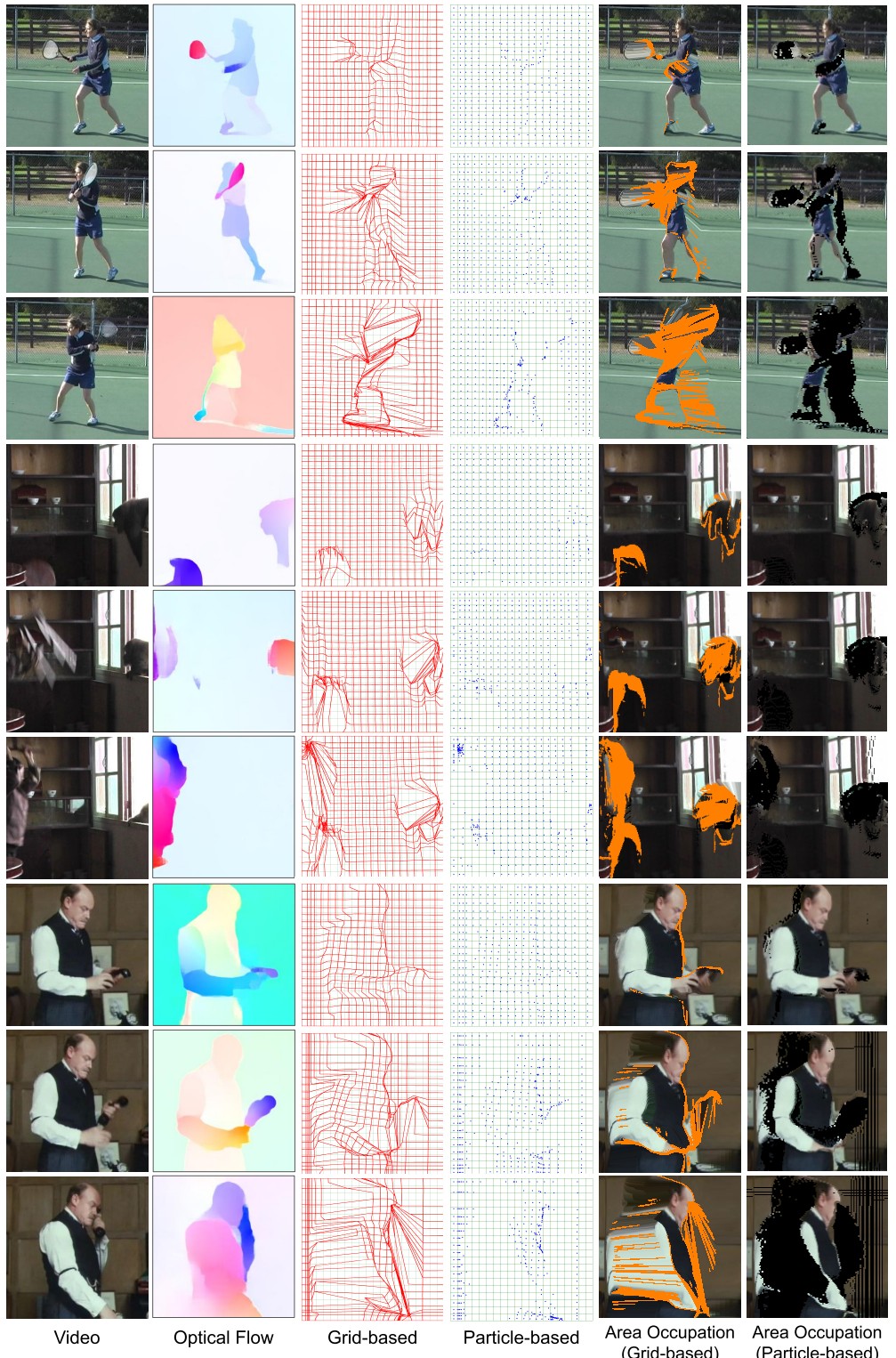

Video    Optical Flow    Grid-based    Particle-based    Area Occupation (Grid-based)    Area Occupation (Particle-based)

Figure B.5: Comparison of grid-based and particle-based variants under non-diffeomorphic deformation maps typically found in real-world applications. The orange pixels are the invalid pixels where area contention occurs. Flow maps are downsampled $10\times$ for better visualization. Image sequence comes from Brox & Malik (2011) while optical flow is computed via Teed & Deng (2020).

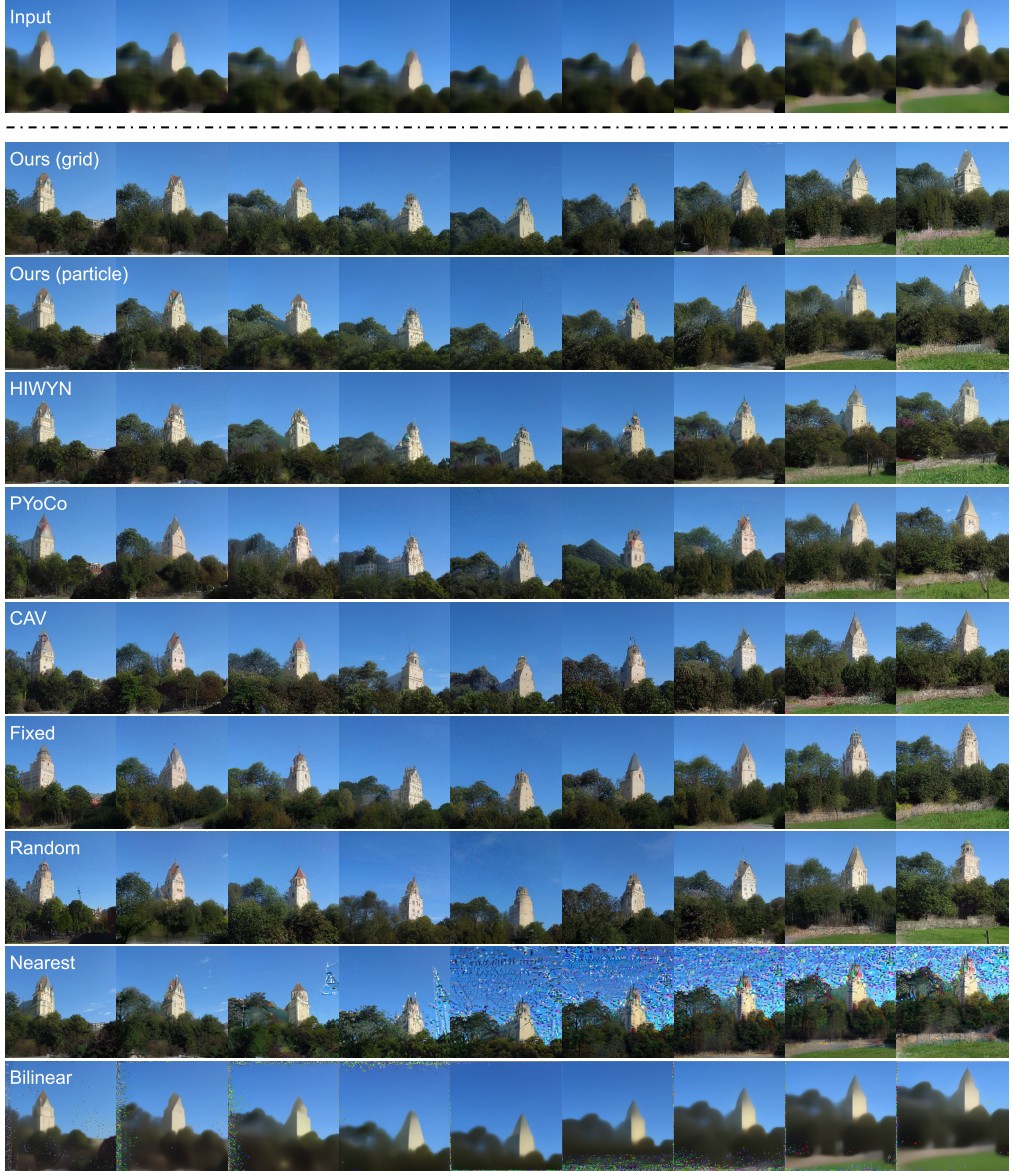

Figure B.6: The full results generated by all the compared methods on the church scene. The interpolation baselines yield noticeably corrupted results, and others yield similar quality *on the image level*. The difference lies in how the details are preserved across frames. We refer to our supplementary video for better visualization of these results.

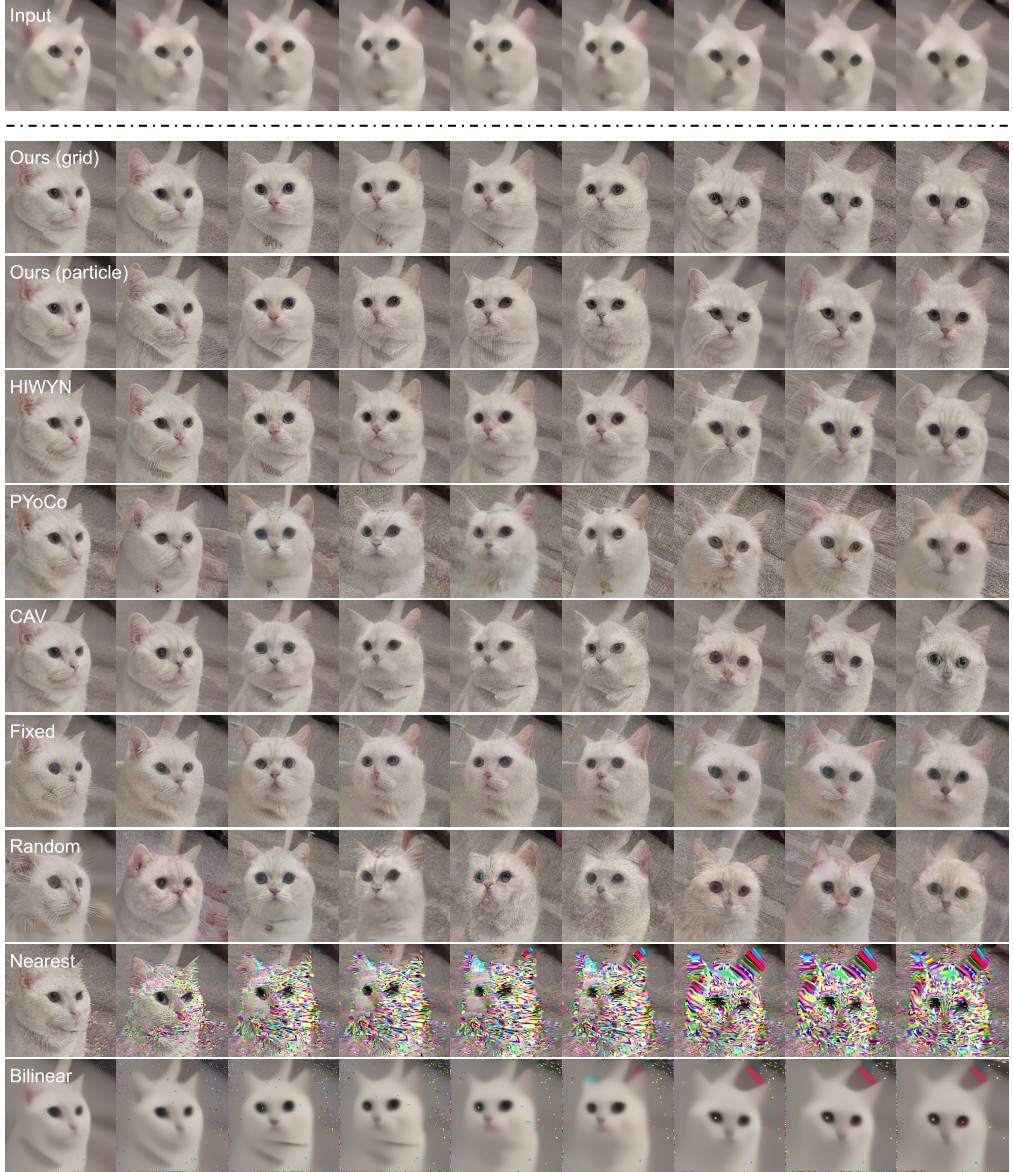

Figure B.7: The full results generated by all the compared methods on the cat scene. We observe that HIWYN and our method (both variants) yield similar results. While both our variants are much faster and more memory-efficient than HIWYN, this makes a particularly strong case for our particle-based variant considering its significantly improved simplicity and efficiency. We refer to our supplementary video for better visualization of these results.

