# OpenReview forum: "Infinite-Resolution Integral Noise Warping for Diffusion Models"
_ICLR.cc/2025/Conference — ICLR 2025 Poster_

### Official Review · Reviewer_XEe5 · 2024-10-30

**Soundness:** 4
**Presentation:** 4
**Contribution:** 3
**Rating:** 6
**Confidence:** 4

**Summary:**

This paper aims to tackle the computational inefficiency of the noise warping algorithm first presented by Chang et al. [1]. To this end, the authors propose an Eulerian perspective of the warping problem, which allows achieving equivalent results with orders of magnitude speed-up. The algorithm scatters increments of Brownian bridges from the original pixels to the warped ones, effectively removing the need for a high-resolution sample, as it simulates the case of infinite resolution. Additionally, the paper proposes a particle-based approach which is simpler to implement, more robust and even faster, at the cost of minor loss in accuracy in real-world scenarios. To support the practical value of the proposed method, the authors show applications with video translation tasks and 3D editing.

[1] How I Warped Your Noise: a Temporally-correlated noise prior for diffusion models. (Chang et al.)

**Strengths:**

1. While the link between noise warping and brownian bridges was noted in previous work, the insights provided by the Eulerian perspective to leverage this connection are novel and theoretically sound.

2. The proposed algorithms greatly improve the computational efficiency of the problem compared to previous work, which can open the door for more potential applications.

3. The particle-based algorithm tackles another limitation of previous work, which was limited to diffeomorphic warpings. This simpler approach can make such method more accessible.

4. The paper is well-written and clear in its explanations, and the proposed method is well validated.

**Weaknesses:**

1. It is hard to tell from the supplementary video results whether noise warping is actually improving coherency, especially in the 3D case.
2. While the noise warping method is now faster, the authors do not show what practical benefits this speed-up can bring (e.g. what new applications does this open the door to), which slightly hinders the significance of this work.
3. As mentioned by the authors, the method is still limited by the accuracy of the provided flow, and the effects of noise warping in video translation tasks remain based on empirical observations.

**Questions:**

Not many questions, but I’m curious to know if the authors have any idea what practical use cases could actually benefit from the almost real-time version of noise warping they proposed?

---

> ### Author Response · Authors · 2024-11-23
> **Response to Reviewer XEe5**
>
> We thank Reviewer XEe5 for their insightful feedback. The reviewer points out that from the supplementary video results, it is hard to tell whether noise warping is actually improving the coherency, especially in the 3D case. The reviewer also notes that our paper does not show the practical benefits that our speed-up can bring (e.g. what new applications does this open the door to). We will address these questions and concerns below.
>
> ### The improvement in coherency is not clear, especially in 3D case
> We believe that **the new results in our updated supplementary video can help elucidate this improvement**. Visual incoherency is **caused jointly by flickers in the structure and the appearance**. Since noise warping excels at handling the high-frequency details, but not the low-frequency structures, the original result (where we use noise warping as the only consistency technique) reduces flickers on the appearance level, but passes through the flickers on the structural level. This can cause the output video to still appear flickery overall, making the improvement less visible.
>
> To account for this, we have integrated our continuous noise warping with I$^2$SB [1], which is a structure-preserving bridge model, and conducted video 4x super-resolution on 4 different scenes. Because I$^2$SB suppresses the structural flickers on its own, when we combine it with our noise warping to suppress the appearance flickers, we achieve much better overall coherency. Importantly, we verify that if we use I$^2$SB **without noise warping**, we either get 1) strong **appearance flickers** with random noise per-frame, or 2) visible **sticking artifacts** with fixed noise. We believe that these **new comparisons vividly illustrate the improved temporal coherency that noise warping brings**.
> * For the 3D case, while developing a compelling use case of volumetric noise warping in 3D applications is an exciting direction for future work that is outside the scope of our paper, we believe that our particle-based variant directly and **uniquely enables this line of research**, which would have been computationally intractable for the existing upsampling-based technique (especially from the memory standpoint). To the best of our knowledge, ours is the first method to achieve distribution-preserving warping of volumetric white noise, and we plan on investigating the utility of such noise in the future.
>
> ### What are the practical benefits of making noise warping faster?
> While in a standard denoising diffusion setting, a full inference pass (with many denoising iterations) indeed requires just a single noise warping operation, there are many other methods that **require noise warping to be computed at each iteration**, making it the bottleneck. For example:
> * **Noise warping + Bridge Models (e.g. I$^2$SB)**
>     * Because bridge models do not use Gaussian noise as initialization, but rather inject noise iteratively during the transport process, **combining noise warping with bridge models requires one noise warp per iteration**. We have validated that using the same copy of warped noise for all iterations breaks the model's generation capabilities. This makes the cost for noise warping a significant bottleneck: on our laptop with a 3070 Ti GPU, one I$^2$SB iteration at 256x256 resolution requires <0.16s per iteration, while warping a 256x256 noise image, as reported in HIWYN [2], requires > 0.6s.
>     * Since we reduce the time cost of HIWYN by 41.7x, the overhead for warping the noise for I$^2$SB becomes **negligible when using our method**. For HIWYN, warping the noise would cost ~4x as much as running I$^2$SB itself, and **a video generated in ~30 mins would require ~2 hours of noise warping**. Since I$^2$SB + noise warping generates the strongest video results for both our paper and HIWYN, the time saving that we can achieve here is significant.
> * **Noise warping + Score Distillation Sampling**
>     * In a similar vein, **SDS also requires one noise warping each iteration**, which makes HIWYN a highly costly option. This is noted by Kwak et. al 2024 [3]. Combining our noise warping with SDS is hence a viable and promising avenue.
>
> We will add this discussion to the revised paper to strengthen the significance of our proposed method.
>
> **References**
>
> [1] Liu, Guan-Horng, et al. "I SB: Image-to-Image Schr" odinger Bridge." arXiv preprint arXiv:2302.05872 (2023).
>
> [2] Chang, Pascal, et al. "How I Warped Your Noise: a Temporally-Correlated Noise Prior for Diffusion Models." The Twelfth International Conference on Learning Representations. 2024.
>
> [3] Kwak, Min-Seop, et al. "Geometry-Aware Score Distillation via 3D Consistent Noising and Gradient Consistency Modeling." arXiv preprint arXiv:2406.16695 (2024).

---

> > ### Comment · Reviewer_XEe5 · 2024-11-24
> > **Thank you for the rebuttal**
> >
> > I thank the authors for their thorough reply, the authors have answered my questions satisfyingly. I thus keep my positive score on the paper.

---

> > > ### Author Response · Authors · 2024-11-27
> > > **Response to Reviewer XEe5's reply**
> > >
> > > We thank Reviewer XEe5 for their response. Inspired by the reviewer’s questions regarding 1) the practical benefits of the speed-up, and 2) the remaining inconsistencies in our results, we have carefully revised our manuscript (uploaded with changes highlighted in blue). In particular, we have added two discussion paragraphs to Section 5 to address these questions in depth. Furthermore, to validate the claim that
> > > > ... the overhead for warping the noise for I$^2$SB becomes negligible when using our method
> > >
> > > as we stated in the rebuttal, we perform an additional timing experiment which shows that our method prepares the noise needed for I$^2$SB in 40.6s (reduced from > 80 minutes with HIWYN), which is indeed very small compared to the ~24 minutes time cost for running the I$^2$SB model. We have included this statistic in the added discussion.
> > >
> > > Since the reviewer has indicated that all questions have been satisfactorily addressed, we would sincerely appreciate it if they could consider raising the score. Please let us know if there are further reservations, and we are committed to improving our paper and providing additional clarifications.

---

### Official Review · Reviewer_gG1q · 2024-10-31

**Soundness:** 4
**Presentation:** 2
**Contribution:** 4
**Rating:** 8
**Confidence:** 3

**Summary:**

The paper presents an advanced noise manipulation method to enhance temporal consistency in video generation using diffusion models. The authors address the computational inefficiency in Chang et al.'s (2024) method, which ensures temporal consistency but requires high-resolution upsampling. By shifting from a Lagrangian to an Eulerian perspective and using Brownian bridge increments instead of high-resolution upsampling, the approach achieves the same precision while significantly improving speed and memory efficiency. The author also introduces two methods for calculating noise overlap areas—grid-based and particle-based. The particle-based approach is faster, more memory-efficient, and more robust to non-injective mappings. Results show that this method not only preserves Gaussian noise properties but also achieves speed improvements by several orders of magnitude, maintaining or even improving temporal consistency in video applications, making it more practical for real-world scenarios.

**Strengths:**

- By using Brownian bridge increments instead of high-resolution upsampling, the proposed method achieves efficient noise manipulation without sacrificing accuracy, greatly enhancing speed and memory efficiency.
- The method maintains temporal consistency of noise in video generation, avoiding flickering or inconsistencies between frames seen in traditional methods, resulting in smoother and more natural video outputs.
- The particle-based noise overlap calculation method is not only faster but also more robust to non-injective mappings, making it suitable for complex, non-linear real-world scenarios and offering high application potential.

**Weaknesses:**

See Questions.

**Questions:**

1. The symbols are not clearly defined, for example, in the description of the Eulerian Procedure on lines 158-166, $M, m_k, l_k ...$ which can lead to difficulties in reading and understanding.

2. The ultimate computation goal is the same as in HIWYN: $ \tilde I_{i,j}$ , as shown in Figure 1, the goal is to obtain the "area" of the red region $ \tilde A_{i,j}$ . However, from the Eulerian View, it is evident that $ \tilde A_{i,j} $ will be dispersed across each $ A_{i,j}$ ; in the example in Figure 2, this accounts for 39%. $ I_{i,j}$ can be considered a Brownian bridge from 0 to $ I_{i,j} $, and the area of each deformed region can be viewed as "time." Using this, it might be possible to directly sample the 39% value of the Brownian bridge and then gather it with other red parts across various $ A_{i,j} $ regions to obtain the final $ \tilde A_{i,j}$ . I am not sure if my understanding is correct. If this understanding is correct, then the differently colored regions within $A_{i,j}$ should originate from sampling with different conditions (as indicated by equation (1)), meaning that each color belongs to a different distribution. Would it then be valid to interpret this as an addition of values from Brownian bridges?

---

> ### Author Response · Authors · 2024-11-23
> **Response to Reviewer gG1q**
>
> We thank Reviewer gG1q for their insightful feedback. The reviewer points out that some of our symbols are not clearly defined, and raises a question regarding our algorithm design. We will address these questions and concerns in the following passage.
>
> ### Undefined symbols
> We thank the reviewer for pointing this out. In the revision, we will clearly define the symbols $M$, $m_k$, $l_k$ before using them in the description of the Eulerian procedure. We will also conduct another comprehensive proof-reading pass to make sure that all our symbols are clearly defined, to ensure a smooth reading experience.
>
> > The ultimate computation goal is the same as in HIWYN: $\tilde I_{i,j}$ , as shown in Figure 1, the goal is to obtain the "area" of the red region $\tilde A_{i,j}$. However, from the Eulerian View, it is evident that $\tilde A_{i,j}$ will be dispersed across each $A_{i,j}$; in the example in Figure 2, this accounts for 39%. $I_{i,j}$ can be considered a Brownian bridge from 0 to $I_{i,j}$, and the area of each deformed region can be viewed as "time." Using this, it might be possible to directly sample the 39% value of the Brownian bridge and then gather it with other red parts across various  regions to obtain the final $\tilde A_{i,j}$. I am not sure if my understanding is correct. If this understanding is correct, then the differently colored regions within $A_{i,j}$ should originate from sampling with different conditions (as indicated by equation (1)), meaning that each color belongs to a different distribution. Would it then be valid to interpret this as an addition of values from Brownian bridges?
>
> The reviewer’s understanding of our method appears correct to us, and adding together increments of multiple Brownian bridges is exactly the key component of our algorithm. The question that we think the reviewer is getting at is this: “As a deformed pixel, when I calculate that I have a 39% area overlap with an undeformed pixel, why can’t I directly **gather** that 39% portion from the undeformed pixel’s Brownian bridge --- but instead need to first store it in a "partition record", and wait for the undeformed pixel to **scatter** the 39% to me in a second pass?”
>
> This is a great question, and we answer it from two aspects:
> * The first aspect is about performance and parallelizability. Because the sampling of Brownian bridges is autoregressive, later samples depend on the outcome of previous ones. Programmatically, this cannot (as far as we know) be done using atomic operations only and would require mutex, which makes it unfriendly to GPU execution. In contrast, if we first **record and then scatter, then both can be done using only atomic operations**, making the program much faster on GPU.
> * Secondly, in our particle-based variant, the "39%" cannot be directly calculated. Rather, deformed pixels only know proximity scores calculated by the distance-based kernels, which need to be **normalized by the undeformed pixel to obtain the area percentage**. So even without considering performance, we cannot perform direct gathering with the particle-based variant.
>
> Please let us know if we have understood your question properly, and we are happy to discuss further.
>
> Finally, we also invite the reviewer to check out our **updated supplementary** video where we combine our method with **I$^2$SB** [1] and **cross-frame attention** [2] to achieve significantly higher levels of visual consistency.
>
> **References**
>
> [1] Liu, Guan-Horng, et al. "I SB: Image-to-Image Schr" odinger Bridge." arXiv preprint arXiv:2302.05872 (2023).
>
> [2] Ceylan, Duygu, Chun-Hao P. Huang, and Niloy J. Mitra. "Pix2video: Video editing using image diffusion." Proceedings of the IEEE/CVF International Conference on Computer Vision. 2023.

---

> > ### Comment · Reviewer_gG1q · 2024-11-25
> >
> > Thanks for the authors' reply. The authors have addressed my concerns. This is a highly insightful approach. I hope the authors can include more details on methods and formulas in Section 2 in the next version. In addition to the previously mentioned symbols, more clarity on $u$, $v$, and $w$ ... in Algorithms 2 and 3 would greatly enhance readability. I would be happy to raise my score.

---

> > > ### Author Response · Authors · 2024-11-28
> > > **Response to Reviewer gG1q's reply**
> > >
> > > We thank Reviewer gG1q for their response. Following the reviewer’s pointers and suggestions, we have revised our manuscript (uploaded with changes highlighted in blue). In particular, we defined the symbols $M$, $l_k$, and $m_k$  before using them in the description of the Eulerian procedure (bottom of page 3), and we added more clarity to the symbols in Algorithms 2 and 3 (page 6). Since $w$ is not related to $u, v$, we have changed $w$ to $\alpha$ to avoid confusion. Also, since $u, v$ in Algorithm 2 and $x’, y’$ in Algorithm 3 represent the same concept (undeformed pixel regions), we have unified them both to $u,v$ for better consistency. We have added a sentence to establish this at the bottom of page 5.
> > >
> > > Please feel free to let us know if there are further details and clarifications that can help improve the exposition. We thank the reviewer again for their insightful input.

---

### Official Review · Reviewer_p4MX · 2024-11-01

**Soundness:** 4
**Presentation:** 4
**Contribution:** 3
**Rating:** 8
**Confidence:** 4

**Summary:**

The paper proposed a different formulation for the "integral noise warping" approach of Chang et al. (2024). The authors prove the relation of this formulation (in infinite resolution) to Brownian bridges, and use it to derive novel implementations for the integral noise warping that do not require upsampling, and thus significantly reduce the computational cost of the method proposed by Chang et al. (2024).

**Strengths:**

- The new interpretation/formulation of integral noise warping is novel and leads to theoretically backed relation to Brownian bridges, whose Markovian property allows to derive convenient implementation methods.

- The derived methods are useful in practical applications where image-based diffusion models are used for video generation in a training-free manner.

- The paper is well-written.

**Weaknesses:**

- There is still inconsistency in the details in the frames.

- The quantitative SDEdit-based video results of the method (and HIWYN) are similar to the competitors.

**Questions:**

- Since integral noise warping does not prevent noticeable inconsistency in the details in the frames, is it possible to enhance your method by adding an explicit temporal consistency term between frames?

---

> ### Author Response · Authors · 2024-11-23
> **Response to Reviewer p4MX**
>
> We thank Reviewer p4MX for their insightful feedback. The reviewer notes that there are still inconsistencies among frames, and raises the question of whether explicit temporal consistency terms can be added to enhance our method. Also, the reviewer points out that the quantitative results of our method (and HIWYN [1]) are similar to those of the compared methods. We will address these questions and concerns below.
>
> ### Why still inconsistent?
>
> * We highlight that the SDEdit videos that we originally showed are generated by **using noise warping as the only temporal consistency technique**. Because **noise warping excels at controlling high-frequency details and textures**, to achieve the best consistency level, it is meant to be **coupled with other techniques that preserve low-frequency structure** and composition. Our original SDEdit experiment deliberately **does not couple** with structure-preserving techniques to **disentangle the impact of noise warping** for illustrative purposes.
> * **In our newly added experiments (as shown in our updated supplementary video)**, we validate this point by showing how our noise warping can readily integrate with existing structure-preserving methods to **achieve much better consistency**.
>     * We showcase the integration with I$^2$SB [2], which is a structure-preserving bridge model, on performing 4x video super-resolution. Because I$^2$SB preserves the low-level structures, the video consistency is far better than our previous SDEdit examples. Meanwhile, we also show that even with I$^2$SB, the **high-frequency preservation achieved by noise warping is crucial to the final consistency**. Testing on 4 different videos, we show that 1) noise warping techniques provide a clear consistency advantage over baselines, and 2) both our variants perform on the same level as HIWYN while being significantly more efficient.
>     * We also showcase the cat and church examples **with added cross-frame attention** in our updated video. It can be observed that the visual stability improves considerably.
> * To sum up, our noise warping method excels at handling high-frequency details and texture, and can be readily combined with other structure-preserving techniques for best performance.
>
> > “Quantitative SDEdit-based video results of our method (and HIWYN) are similar to the competitors”
>
> * We agree with this observation, and we point out that these quantitative metrics have limitations and cannot holistically reflect the desired performance. For example, the KID score for measuring realism takes diversity into account, which means that more jittery videos tend to receive higher scores. On the other hand, the $L_2$ score measuring faithfulness will penalize deviation from the condition image, which means videos without details tend to receive higher scores. So far, we still find the video quality to be most reliably gauged visually, and the visual benefits of our method is made clear in our **updated supplementary video**.
>
> **References**
>
> [1] Chang, Pascal, et al. "How I Warped Your Noise: a Temporally-Correlated Noise Prior for Diffusion Models." The Twelfth International Conference on Learning Representations. 2024.
>
> [2] Liu, Guan-Horng, et al. "I  SB: Image-to-Image Schr" odinger Bridge." arXiv preprint arXiv:2302.05872 (2023).

---

> > ### Comment · Reviewer_p4MX · 2024-11-24
> >
> > I have read the authors response and keep my positive view of this paper.

---

> > > ### Author Response · Authors · 2024-11-28
> > >
> > > We thank Reviewer p4MX again for the review.

---

### Official Review · Reviewer_jdjC · 2024-11-11

**Soundness:** 3
**Presentation:** 4
**Contribution:** 2
**Rating:** 3
**Confidence:** 3

**Summary:**

The topic of the paper is to develop computationally efficient algorithms to use image diffusion models temporally, i.e. for generating consistent videos or solving video inverse problems. The authors innovate on the noise-warping mechanism and they propose a computationally efficient method to transform the noise between frames that preserves the Gaussian prior distribution. The authors experimentally validate their method for video super-resolution and 3-D problems.

**Strengths:**

1) The paper is very well written and the ideas are presented in a natural way.

2) The method is significantly faster compared to the baseline.

3) The fuzzy variant of the method is very clever: significant computation time is saved and the method is more robust as evaluated in real-world applications.

4) The ideas presented are sound and the Euclidean View offers an alternative way to think about the problem of noise warping that I enjoyed reading about.

**Weaknesses:**

1) The biggest weakness of this paper is that the experimental results are very weak. I downloaded the supplementary material and looked at the videos; the quality is unsatisfactory. While the proposed method offers more consistency than the existing baselines, it is still very inconsistent. The extent of inconsistency is so high, that I can't think of any applications that would leverage this method.

2) At the same time, I am not very confident that the considered approach is meaningful. It is true that at the moment there are many powerful image diffusion models and not so many video diffusion models. However, the current level of quality of solving inverse problems with image diffusion models appears to be very low, judging from the results of this paper. I am not sure if this is a fundamental limitation of the approach or whether it will be resolved in the future. But also, as time goes by, more and more powerful video diffusion models are becoming available that can be used as priors for solving temporal inverse problems.

3) I believe that one of the reasons the experimental results are not that strong is the equivariance of the network itself. Even if the input noise transformations are properly handled, maybe the network itself does not provide equivariant predictions which could cause the creation of flickering and inconsistencies. This is argued in this work: https://arxiv.org/pdf/2410.16152.

4) I also believe that the paper would benefit from more experiments on other inverse problems. Since a pre-trained prior is available, one could experiment with other types of problems such as inpainting or even denoising as the image follows simple geometrical transformations (zoom-in/out, shifts, etc).

**Questions:**

See also weaknesses above.

1) The experimental results in the website of the How I Warped Your Noise paper (https://warpyournoise.github.io/), seem of much higher consistency compared to the ones provided in the zip file by the authors. Could the authors comment on this? Were there any reproducibility issues?

2) This [paper](https://arxiv.org/pdf/2410.16152) claims that "white noise is not compatible with the idea of using a generative model to interpolate deformed functions". Could the authors comment on this and the preservation of the Gaussian prior under different transformations?

3) Could the authors investigate if having an equivariant generator improves the experimental results?

---

> ### Author Response · Authors · 2024-11-22
> **Response to Reviewer jdjC [1/3]**
>
> We thank Reviewer jdjC for their insightful feedback. The reviewer raises a concern regarding the quality of our result videos. While we agree that the consistency of these results is worse than that of the videos on HIWYN’s website, we believe this discrepancy reduces neither the practical nor theoretical impact of our method, as we would clarify this misunderstanding and **present several new experimental results** to further validate our claims. **We refer the reviewer to our updated supplementary video to see our newly added results.**
>
> First, we point out that the frame-wise consistency in the results of How I Warped Your Noise (HIWYN) [1] is achieved by the joint effort of multiple orthogonal techniques. Certain techniques, such as cross-frame attention and feature-map injection, handle low-frequency structural compositions, while noise warping excels at high-frequency details and textures. The **results that we originally included deliberately avoid integrating other structure-preserving techniques** for the following reasons.
>
> * Our paper is set to do one thing and one thing well — to drastically speed up HIWYN without quality compromise. On one hand, **with the theoretical connection we establish, we are guaranteed that our method won’t perform worse than HIWYN**. On the other hand, our accuracy advantage over HIWYN in theory is not visually significant in practice (with HIWYN using a high enough upsampling resolution). We find these two points to together imply that our method, when used as the drop-in replacement to HIWYN, will yield the same level of visual quality.
> * Regarding this point, while it would be most persuasive to directly recreate HIWYN’s experiments using our continuous noise warping, we point out that **after reaching out to the authors of HIWYN for source code (which is not open-source), they did not share it with us**, which makes "apples-to-apples" comparisons difficult. While their core method is clearly explained, they **omitted many key details for reproducing the visual results**. To mention a few:
>     * In their SDEdit (bedroom) example, the $t_0$ parameter is not provided. The uncompressed input images and the optical flow are also not available. We have validated that the optical flow generated using RAFT yields far inferior quality.
>     * In their I$^2$SB example, some necessary configurations (e.g. nfe parameter) are not specified, and the upsampling type is also not specified (pooling vs. bicubic).
> * Finally, considering both of the above points, at the time of submission we decided that the value of reproducing such apples-to-apples tests is outweighed by the difficulty. Taking a different direction, we use our experiments to **show the impact of noise warping in the absence of other touch-up techniques**. As noted in the HIWYN paper (bottom of p.32), integrating other techniques can overshadow the impact of the noise. As a result, we design our SDEdit experiment without cross-frame attention (or synthetic data), to expose the behaviors of noise warping in a stress test. Our results both 1) confirm that noise manipulation on its own can already make a clear difference to vanilla image models, and 2) expose what can be controlled by warped noise and what cannot – for instance, in the church scene, the tree’s texture can be controlled, while the tower’s structure cannot, which offers valuable insight about the noise warping technique that has not been showcased so far.
>
> **[To be continued: 1/3]**

---

> ### Author Response · Authors · 2024-11-22
> **Response to Reviewer jdjC [2/3]**
>
> With that said, we also agree with Reviewer jdjC that, despite the theoretical guarantee, it is also important to experimentally validate that our new method can yield high-quality results when integrated into a more advanced pipeline, to confirm its real-world usability. We have revised our submission focusing on this point. **In our updated supplementary video, we add several new experiments** conducted during the rebuttal period to showcase successful integration with additional techniques. In particular:
>
> * We have integrated our continuous noise warping with I$^2$SB [2], and tested the 4x super-resolution task with side-by-side comparisons with HIWYN. Because I$^2$SB is a **structure-preserving** bridge method, we **achieve far better visual stability than the SDEdit results we showed previously**.
>     * Since the key details of I$^2$SB integration were not provided in the HIWYN paper, we do our best to tune our integrated system so that our reimplementation of HIWYN generates results as close as possible to the results on [their website](https://warpyournoise.github.io).
>         * As shown in our updated supplementary video, the stone wall in their website’s "bear" video appears slightly more stable and more blurry than our reimplementation. Also, our reimplementation does not generate the rainbow-like artifact trailing the bear. Overall, both videos show a similar level of visual consistency.
>     * We reproduce the bear example with both our method variants, and compare against HIWYN along with random and fixed noise baselines. We validate that **both variants perform on the same level as HIWYN while being much more computationally efficient**.
>     * We also show additional I$^2$SB comparisons on three more complex scenes from the DAVIS dataset to highlight the robustness and versatility of our method.
>
> * We have coupled **SDEdit with cross-frame attention** a la Pix2Video [3] (anchoring every 3 frames) and re-ran our church and cat experiments. We also reduce the $t_0$ parameter in SDEdit from 0.6 to 0.4 to preserve more structure from the input video. As shown in our updated supplementary video, these two changes are enough to **significantly suppress the flickers in the generated results**.
>     * Here we also highlight that our inputs are real-world videos with estimated optical flows with RAFT [4], as opposed to the synthetic scenes in HIWYN.
>
> **[To be continued: 2/3]**

---

> ### Author Response · Authors · 2024-11-22
> **Response to Reviewer jdjC [3/3]**
>
> ## Image-based vs. Video-based
> We believe that the importance of image-based video generation techniques will be persistent. In addition to the reasons already mentioned in the paper, we will add the following points in the revision to elaborate.
> * First, we note that image learning is a strictly easier task than video learning. So we foresee the single-frame quality gap between image and video models to remain as SOTA develops. This persistent gap will keep adapters like this work relevant.
> * Secondly, although we demonstrate image-to-video with temporal consistency, the method can be directly used to encourage other types of consistency, e.g., 3D consistency, which extends beyond video generation.
>
> ## Discussion with Warped Diffusion [5]
> We agree that the Warped Diffusion paper addresses the same consistency problem as ours and provides profound insights into network equivariance. Here, we find it important to emphasize that, **to achieve equivariance, Warped Diffusion actually evokes noise warping as a core subroutine (see line 5 of its Alg. 1)**; and our proposed continuous noise warping algorithm offers an attractive new way of realizing this subroutine.
> * We explain why **our method is readily compatible with and potentially desirable for Warped Diffusion**. As discussed in the final paragraph of Warped Diffusion, Appendix C, their goal is to compute $\xi(T^{-1}(E_k)) | \xi(E_k)$, which **can be achieved both by integral noise warping (HIWYN and ours) or RFF projection**. They picked RFF projection over HIWYN for numerical considerations, and **cited RFF's 1) infinite resolution and 2) computation efficiency as two main reasons (see last sentence of section 3.1), which are precisely our main improvements over HIWYN**. Hence, we postulate that using our continuous noise warping in place of RFF projection in Warped Diffusion might be desirable with two potential benefits:
>     * Their RFF-based noise warping generates **correlated noise instead of white noise** (see their Figure 6). **Because off-the-shelf models are usually pre-trained with white noise, this creates a domain gap that requires additional fine-tuning** (see last paragraph of section 4.1). By warping white noise directly, our method circumvents this domain gap.
>     * Since they **report their RFF-based noise warping to be 16× faster than HIWYN while ours is 41.7x faster**, using our method promises to provide a significant further speed-up.
>
> > “We show in Section 2.3 that a white noise process is not compatible with the idea of using a generative model to interpolate deformed functions”
> * As explained in Warped Diffusion Section 2.3, the "white noise process" mentioned here corresponds to the "random noise" baseline in our experiments, where each noise value is individually drawn. This lack of smoothness breaks the notion of interpolation, therefore making it "incompatible". To address this, one needs to use some form of "noise warping" to inject correlation and smoothness informed by the deformation $T$, where the options are either stochastic interpolation (HIWYN and our method) or the RFF projection method that they propose. So this remark in the Warped Diffusion paper precisely emphasizes the importance of noise warping.
>
> We agree with the reviewer that the lack of equivariance in the network would be a source of visual inconsistency in our demo videos. Meanwhile, we note that the advent of Warped Diffusion precisely illustrates why **the value of our paper should not be dictated by its demo video quality. After all, our core contribution is a theoretically rigorous and highly performant solution to an important low-level problem (noise warping), a solution that higher-level approaches like Warped Diffusion can build upon to achieve SOTA results**.
>
> **References**
>
> [1] Chang, Pascal, et al. "How I Warped Your Noise: a Temporally-Correlated Noise Prior for Diffusion Models." The Twelfth International Conference on Learning Representations. 2024.
>
> [2] Liu, Guan-Horng, et al. "I $^ 2$ SB: Image-to-Image Schr\" odinger Bridge." arXiv preprint arXiv:2302.05872 (2023).
>
> [3] Ceylan, Duygu, Chun-Hao P. Huang, and Niloy J. Mitra. "Pix2video: Video editing using image diffusion." Proceedings of the IEEE/CVF International Conference on Computer Vision. 2023.
>
> [4] Teed, Zachary, and Jia Deng. "Raft: Recurrent all-pairs field transforms for optical flow." Computer Vision–ECCV 2020: 16th European Conference, Glasgow, UK, August 23–28, 2020, Proceedings, Part II 16. Springer International Publishing, 2020.
>
> [5] Daras, Giannis, et al. "Warped Diffusion: Solving Video Inverse Problems with Image Diffusion Models." arXiv preprint arXiv:2410.16152 (2024).

---

> > ### Comment · Reviewer_jdjC · 2024-11-23
> > **Thank you for your rebuttal**
> >
> > Dear authors,
> > Even though I truly appreciate your heroic efforts during the rebuttal, I still find the achieved results unsatisfactory, and I still do not understand who the target audience of this work is.
> >
> > The paper itself is positioned as a practical advancement: it provides a computationally efficient (and still theoretically rigorous) algorithm for noise warping. In my opinion, the quality of the results is low; hence, it does not matter if it runs fast or not because it does not seem useful for practitioners anyway. To put it concretely, the practical value of this method seems very low and hence at this point, I believe that (in terms of practical impact) the priority should be to make it work better, not faster.
> >
> > Given the previous point, I find the practical significance of the proposed algorithm low. While I did enjoy the theoretical aspects of this work, I also do not think that the theoretical contributions are enough to merit publishing this work with such low-quality experimental results.
> >
> >
> > Once again, I appreciate your rebuttal efforts, but unfortunately, I cannot recommend acceptance.

---

> ### Author Response · Authors · 2024-11-23
> **Response to Reviewer jdjC's reply**
>
> We thank Reviewer jdjC for their prompt and courteous reply. We understand the reviewer's main point as: "the low quality of our results renders the method's efficiency and theory irrelevant".
> > ... the quality of the results is low; hence, it does not matter if it runs fast or not because it does not seem useful for practitioners anyway
>
> While the logic is valid, we respectfully disagree that it applies to our case. We believe this arises from a **misunderstanding of our project's scope**. We will explain why:
> * In our project, we take as input flow maps $F$, use our continuous noise warper $\mathcal{W}$ to generate deforming noise images $\mathcal{W}(F)$, and feed them into an image model $\mathcal{I}$ (SDEdit, I$^2$SB, etc.) to generate output videos $\mathcal{I}(\mathcal{W}(F))$.
> * We propose $\mathcal{W}$ as a state-of-the-art method for **noise warping**, but **not** $\mathcal{I}\circ\mathcal{W}$ as a novel **video generation** method. This has been made clear in our paper.
> * In the scope of noise warping, because $\mathcal{W}$ is infinite-resolution, **it achieves the highest possible quality**.
>
> When the reviewer mentions **"low-quality results", do they refer to $\mathcal{I}(\mathcal{W}(F))$ or $\mathcal{W}(F)$?** As we propose $\mathcal{W}$ as a noise warping innovation, our method should be judged by the quality of $\mathcal{W}(F)$ instead of $\mathcal{I}(\mathcal{W}(F))$ .
>
> * **Indeed, if we were to** put forth $\mathcal{I} \circ \mathcal{W}$ as a video generation method, then we should be **comparing $\mathcal{I} \circ \mathcal{W}$ with the video generation SOTA**. In that case, we agree with the reviewer that this $\mathcal{I} \circ \mathcal{W}$ is unsatisfactory.
> * However, because we are **in fact** proposing $\mathcal{W}$ as a noise warping method, we would instead **compare $\mathcal{W}$ with the SOTA method in noise warping, which is HIWYN**.
>     * In our paper, we compare with HIWYN from three aspects: (A) theoretical connection, (B) numerical validation (Gaussianity, independence, convergence), and (C) video generation quality.
>         * Note that **we build $\mathcal{I} \circ \mathcal{W}$ as a way to facilitate (C) -- visual comparisons between our $\mathcal{W}$ and HIWYN**. Specifically, we do not propose $\mathcal{I} \circ \mathcal{W}$ as our new video generation technique.
>     * Based on our original and rebuttal experiments, we **conclude that our generated video quality is on par with HIWYN**, which satisfies (C). Given (A), (B) and (C), we claim that our noise warper $\mathcal{W}$ achieves **SOTA quality in noise warping** while being drastically faster, therefore offering an ideal solution **to the noise warping problem.**
> * Because **we only claims the superiority of $\mathcal{W}$ in terms of noise warping**, we only need to compare against HIWYN, the SOTA in noise warping. The fact that $\mathcal{I} \circ \mathcal{W}$ is sub-par to **video generation SOTA** is not relevant to this claim.
>
> Once we establish our contribution **in the scope of noise warping**, the next important question to ask is: **"If SOTA noise warping cannot translate to high video generation quality, then why should one care?"** Here we make the following points:
> * Noise warping is a low-level component that needs to be integrated in a higher-level system to generate videos. While our implementation of this integrated system is vanilla, **there already is strong evidence that noise warping is a key component to advanced systems that yield SOTA video generation quality** [1].
>     * *"A natural question is whether we can omit completely the noise warping scheme since equivariance is forced at inference time. We ... found that omitting the warping significantly deteriorates the results"* [1, pg. 9-10]
> * In addition to video generation, noise warping has also been bridged with 3D generation [2] to yield high fidelity results. This suggests that the ability to **manipulate noise according to pre-specified deformation maps is a general and fundamental need** across various types of generative modeling.
>
> Following these findings by [1] and [2], we foresee an increasing number of future works to consider incorporating noise warping in their systems. In these cases, a question that naturally arises would be: "What is the best tool to perform this noise warping, in terms of accuracy, efficiency, robustness, and 3D extensibility?" Because of the significant, strict, and comprehensive improvements our method makes to the noise-warping technique, it presents itself as a strong (if not ideal) candidate. We believe this offers a counter-argument to Reviewer jdjC's remark that:
> > I can't think of any applications that would leverage this method
>
> **References**
>
> [1] Daras, Giannis, et al. "Warped Diffusion: Solving Video Inverse Problems with Image Diffusion Models." (2024).
>
> [2] Kwak, Min-Seop, et al. "Geometry-Aware Score Distillation via 3D Consistent Noising and Gradient Consistency Modeling." (2024).

---

### Meta-Review · Area_Chair_va2W · 2024-12-19

**Metareview:**

This paper aims to develop computationally efficient algorithms to use image diffusion models. An alternative algorithm of the integral noise warping approach by Chang et al. (2024) is presented. It achieves infinite-resolution accuracy while simultaneously reducing the computational cost by orders of magnitude. After the rebuttal, it receives mixed ratings, including one reject, one borderline accept, and two accept. The strength of the paper, including the clear motivation, interesting ideas, extensive experiments, and good results, are well recognized. I agree with them and think the current manuscript meets the requirements of this top conference. Please also address the Reviewer wNg9's concerns in the revised manuscript.

**Additional Comments On Reviewer Discussion:**

Most of the concerns are well addressed by the authors in the discussion.

---

### Decision · Program_Chairs · 2025-01-22

Accept (Poster)